# One-step automated bioprinting-based method for cumulus-oocyte complex microencapsulation for 3D *in vitro* maturation

**Antonella Mastrorocco**[1¤a]*, **Ludovica Cacopardo**[2], **Nicola Antonio Martino**[1¤b], **Diana Fanelli**[3], **Francesco Camillo**[3], **Elena Ciani**[1], **Bernard A. J. Roelen**[4], **Arti Ahluwalia**[2,5☉], **Maria Elena Dell'Aquila**[1☉]

1 Department of Biosciences, Biotechnologies and Biopharmaceutics, University of Bari Aldo Moro, Bari, Italy, 2 Research Centre E. Piaggio, University of Pisa, Pisa, Italy, 3 Department of Veterinary Sciences, University of Pisa, Pisa, Italy, 4 Department of Clinical Sciences, Embryology, Anatomy and Physiology, Faculty of Veterinary Medicine, Utrecht University, Utrecht, The Netherlands, 5 Department of Information Engineering, University of Pisa, Pisa, Italy

☉ These authors contributed equally to this work.
¤a Current address: Faculty of Veterinary Medicine, University of Teramo, Teramo, Italy
¤b Current address: Department of Veterinary Sciences, University of Torino, Torino, Italy
* antonella.mastrorocco@uniba.it

**Data Availability Statement:** All relevant data are within the manuscript.

## Abstract

Three-dimensional *in vitro* maturation (3D IVM) is a promising approach to improve IVM efficiency as it could prevent cumulus-oocyte complex (COC) flattening and preserve its structural and functional integrity. Methods reported to date have low reproducibility and validation studies are limited. In this study, a bioprinting based production process for generating microbeads containing a COC (COC-microbeads) was optimized and its validity tested in a large animal model (sheep). Alginate microbeads were produced and characterized for size, shape and stability under culture conditions. COC encapsulation had high efficiency and reproducibility and cumulus integrity was preserved. COC-microbeads underwent IVM, with COCs cultured in standard 2D IVM as controls. After IVM, oocytes were analyzed for nuclear chromatin configuration, bioenergetic/oxidative status and transcriptional activity of genes biomarker of mitochondrial activity (*TFAM*, *ATP6*, *ATP8*) and oocyte developmental competence (*KHDC3*, *NLRP5*, *OOEP* and *TLE6*). The 3D system supported oocyte nuclear maturation more efficiently than the 2D control (P<0.05). Ooplasmic mitochondrial activity and reactive oxygen species (ROS) generation ability were increased (P<0.05). Up-regulation of *TFAM*, *ATP6* and *ATP8* and down-regulation of *KHDC3*, *NLRP5* expression were observed in 3D IVM. In conclusion, the new bioprinting method for producing COC-microbeads has high reproducibility and efficiency. Moreover, 3D IVM improves oocyte nuclear maturation and relevant parameters of oocyte cytoplasmic maturation and could be used for clinical and toxicological applications.

**Funding:** This study was funded by UE - FSE-FSER, PON RI 2014-2020 Action I.1 - "Innovative doctoral of industrial interest" - a.a. 2016/2017, XXXII cycle, PhD Program "Functional and Applied Genomics and Proteomics" - DOT1302781" - Grant n°2. A.M. was also supported by the Research Grant n° VERI00086 in the framework of the Project "Demetra" (Dipartimenti di Eccellenza 2018- 22; CUP code: C46C18000530001) funded by the Italian Ministry for Education, University and Research.

**Competing interests:** The authors have declared that no competing interests exist.

## Introduction

The cumulus-oocyte complex (COC) is a multicellular structure characterized by three-dimensional (3D) architecture and bi-directional crosstalk between the oocyte and its surrounding cumulus cells. Paracrine interactions among these cells govern various processes of oogenesis culminating in oocyte meiosis resumption, maturation and ovulation [1–4]. Despite considerable success achieved in Assisted Reproductive Technologies (ARTs) for *in vitro* production of embryos, both in human reproductive medicine [5, 6] and the animal production industry [7–9], current conditions used for *in vitro* oocyte maturation (IVM) are still sub-optimal compared with *in vivo* conditions [6, 10–13].

The majority of ART studies have been conducted in conventional 2D culture systems. This may have led to limitations due to loss of cell native phenotype [14]. Bioengineering approaches have been employed to develop 3D culture methods in different cell systems and, over the last decade, there has been growing interest in cell microencapsulation, a procedure involving the enveloping of cells within a semipermeable polymeric matrix able to mimic the native tissue architecture and physiology. 3D culture methods for IVM after COC microencapsulation have become a goal for researchers in the field of ARTs [15] as microencapsulation may prevent COCs flattening on the bottom of the plate, typical of a traditional 2D culture system, which causes reduction of about 50% of the cell surface exposed to medium [16].

Suitable biomaterials and microfabrication procedures are important for microencapsulation. Among biocompatible materials, alginate hydrogels have been successfully used for culturing and studying different cell types [17] and have been widely employed in long-term *in vitro* culture of ovarian follicles [18–31]. In recent years, IVM culture of COCs in 3D alginate microenvironments has been reported in various animal models with encouraging results [18, 32–36]. Methods for fabricating microbeads containing a COC, namely "COC-microbeads", need to be chosen according to their ability to control microbead shape and dimensions and generate reproducible 3D IVMs. Various bioprinting technologies have been developed [37] and a compact open source tool, the spherical hydrogel generator (Sphyga), was developed for the fabrication of highly reproducible hydrogel-based microbeads with predictable shapes and diameters [38]. This system is able to create a wide range of working conditions for fabricating tailored microbeads and thereby to generate biologically relevant microenvironments for many biomedical applications [38].

Assessing oocyte maturation and acquisition of developmental competence after IVM includes analysis of nuclear and cytoplasmic maturation. Nuclear maturation involves chromosomal segregation, whereas cytoplasmic maturation involves organelle redistribution, cytoskeleton dynamics and molecular maturation [39]. In the context of organelle redistribution, mitochondria position at different locations and increase their activity during the transition from the germinal vesicle stage to the metaphase II stage [39, 40]. Molecular maturation includes transcription, storage and processing of maternal mRNA [39]. Transcription of specific genes, particularly genes coding for proteins involved in mitochondrial functions, can predict specific aspects of oocyte maturation and, expression of *TFAM*, *ATP6* and *ATP8* can reflect oocyte quality [41]. Indeed, *TFAM* encodes a transcription factor that binds mitochondrial DNA [42]. *ATP6* and *ATP8* are mitochondrial genes coding for two subunits of the complex V of the respiratory chain, namely ATP synthase, involved in ATP production [43–45]. The subcortical maternal complex (SCMC) is known to play essential roles in early embryo development and female fertility [46–51]. It was first described in the subcortex of mouse oocytes and preimplantation embryos as a 669–2000 KD molecular weight complex composed of several proteins encoded by maternal effect genes (MEG) [46]. They include *OOEP* (Oocyte Expressed Protein, also known as Factor Located in Oocytes Permitting Embryonic

Development, *FLOPED*), *NLRP5* (NLR family, pyrin domain containing protein 5, also known as Maternal Antigen That Embryo Requires, *MATER*), *TLE6* (Transducin Like Enhancer of Split 6), *KDHC3* (also known as *FILIA*) and zinc finger BED-type containing 3 (*ZBED3*), and plausibly also *NLRP2* and *PADI6* (Peptidyl-Arginine DeIminase 6) [46, 49]. The SCMC is essential for zygote progression beyond the first embryonic cell divisions and is considered as biomarker of oocyte developmental competence [47].

Given these premises, the aims of the present study were: 1) to set up the conditions for generating reproducible COC-microbeads by an alginate-based bioprinting technology; 2) to establish an efficient 3D IVM culture system in the sheep as a large animal model.

## Materials and methods

### Chemicals

All chemicals for *in vitro* cultures and analyses were purchased from Sigma-Aldrich (Milan, Italy) unless otherwise indicated.

### Collection of ovaries

Ovaries from adult sheep (over 1 year) were recovered at local slaughterhouses (Matteoni SNC, Lucca, Italy and Van Kooten Vleesgroothandel B.V., Oeverweg 3, 3417 XK Montfoort, The Netherlands). Animals were subjected to routine veterinary inspection in accordance with the specific health requirements stated in Council Directive 89/556/ECC and subsequent modifications and were slaughtered according to national and EU regulations in full compliance with regulation (EC) n.1099/2009. In all sites, ovaries were transported to the laboratory at room temperature within 4 hours after slaughter.

### COC retrieval

For COC retrieval, ovaries from adult cyclic ewes underwent follicular fluid aspiration using an 18 G needle followed by slicing [52]. Follicular contents were released in sterile Petri dishes containing phosphate buffered saline (PBS) supplemented with 1 mg/mL heparin. Only COCs with at least three intact cumulus cell layers and homogenous cytoplasm were selected for culture [53].

### *In vitro* maturation (IVM) medium

IVM medium composition was TCM-199 medium with Earle's salts buffered with 5.87 mM HEPES and 33.09 mM sodium bicarbonate, supplemented with 0.1 g/L L-glutamine, 2.27 mM sodium pyruvate, calcium lactate pentahydrate (1.62 mM $Ca^{2+}$, 3.9 mM Lactate), 50 μg/mL gentamicin, 20% (v/v) fetal calf serum (FCS), 10 μg/mL ovine follicle stimulating hormone (FSH), 20 μg/mL ovine luteinizing hormone (LH) and 1 μg/mL 17β estradiol [54]. IVM medium and paraffin oil were pre-equilibrated for 1 hour under 5% $CO_2$ in air at 38.5°C. IVM culture was performed in 4-well plates (Nunc Intermed, Roskilde, Denmark). Each culture well contained 400 μL IVM medium and was covered with 400 μL of lightweight paraffin oil to prevent evaporation of culture and bacterial contamination. Prior to microbead or COC upload, plates were equilibrated for 30 minutes under 5% $CO_2$ in air at 38.5°C.

### Alginate microbead fabrication

Spherical alginate-based microbeads with controlled and predictable shape, size and stability, were fabricated using the spherical hydrogel generator Sphyga [38]. It is a compact device composed of an extrusion module (i.e. commercial syringe and linear actuation stage) and

dedicated software which allows the control of the system's working parameters. Through the software, solution flow rate is set and controlled in the range of $(5-20 \pm 1)$ $\mu Ls^{-1}$, while external air pressure $(1-100 \pm 0.4)$ kPa can be applied to the tip of the syringe to accelerate droplet separation. As the system is activated, the plunger moves down the syringe barrel, the solution flows through the needle and a drop, with size depending on the internal diameter of the needle, is generated on its tip. The liquid droplet lands in a beaker filled with a crosslinking solution whereupon it forms a hydrogel bead. In these microbeads, cells are enclosed in a 3D microenvironment with COC-related dimensions (about 1000 $\mu m$ in radius) in a biopolymer whereby an adequate supply of oxygen and nutrients in the media as well as waste removal is guaranteed [55, 56]. Fig 1 shows the spherical hydrogel generator Sphyga (A, B), the calcium-induced gelification (C) and the Sphyga set-up for microbead preparation (D-E). For adaptations to ovine COC structure and size, alginate solutions were prepared in IVM medium as described above and alginate biopolymer was physically crosslinked by using 100 mM $CaCl_2$ in deionized water [15, 38]. All solutions were equilibrated at 38.5°C under 5% $CO_2$ in air for at least 1 hour before use. The flow rate was kept constant at 10 $\mu Ls^{-1}$ without applying external air pressure [57] to reduce the shear stress which could damage COC viability. Briefly, after setting the parameters through the Sphyga graphical user interface (GUI; [38]), the syringe was filled with the alginate solution, the system was activated, and the beads were collected in a beaker containing 100 mL of $CaCl_2$ solution placed at a distance of 5 cm from the needle tip. Newly formed alginate microbeads were kept immersed in the $CaCl_2$ solution for 5 minutes to allow homogeneous crosslink formation. They were then collected, rinsed with fresh IVM medium and characterized for their morphology and stability in culture conditions. In order to produce microbeads of desired size and shape, the use of needles with different internal diameters (610, 508, 406, 330, 152 and 102 $\mu m$) was tested. The effects of two alginate concentrations (1% and 2% w/v) were analyzed (Fig 1F).

## Microbead morphological characterization and stability in culture conditions

Microbead features were characterized as function of fabrication parameters acquiring bright-field images at different time points (immediately after fabrication up to 24 hours) using an inverted microscope (Olympus IX81, Italy). For microbead morphology, images retrieving dimensions (radius) and the shape descriptor "aspect ratio" [58]) were analyzed using the *Analyze Particles* tool (ImageJ [59]). The aspect ratio is defined as the ratio between the major and the minor axis of an ellipse, with a value of 1.0 indicating a circle. In order to assess the long-term stability of alginate microbeads in IVM medium, alginate microbeads were placed in multi-well Petri dishes (Nunc Intermed, Roskilde, Denmark) containing 400 $\mu L$ of IVM medium and maintained for 24 hours at 38.5°C under 5% $CO_2$ in air to simulate the IVM conditions. Bright-field images of the beads were acquired at intervals of 4 hours and microbead radius and aspect ratio were analyzed.

## Alginate COC-microbead generation

For biological studies, alginate microbeads were fabricated by using the parameters (alginate and crosslinking solution concentrations and needle size) selected in the first part of the study. All experiments were carried out with Sphyga positioned under a sterile hood [38]. Sodium alginate solutions were prepared in IVM medium, filtered and kept at 38.5°C under 5% $CO_2$ in air. The alginate biopolymer solution was physically crosslinked using 0.1M $CaCl_2$ at 38.5°C. A needle with appropriate internal diameter was used, maintaining a flow rate of 10 $\mu Ls^{-1}$. After setting the parameters through the Sphyga GUI, the syringe was filled with the

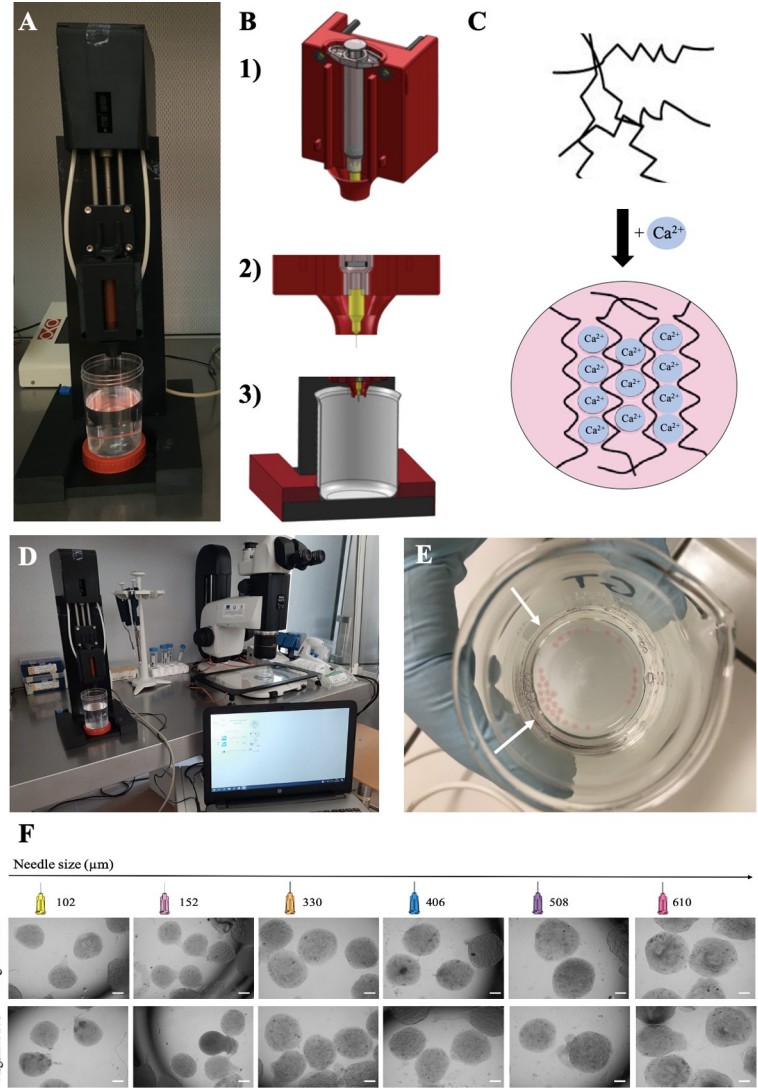

**Fig 1. Spherical hydrogel generator Sphyga set up and microbead preparation procedure.** (A) Extrusion module. (B) Mechanical framework with coaxial design for holding the syringe (B1) and to direct microdroplets into the beaker (B2-3). (C) Ungelled sodium alginate biopolymer fibers followed by calcium-induced alginate gelification. (D) Sphyga set up under a laminar flow hood with dedicated software. (E) Freshly generated alginate microbeads (white arrows) in 100 mM $CaCl_2$ solution. (F) Photomicrographs showing representative images of microbeads obtained with 1% or 2% alginate and needles of different size. Scale bar represents 300 μm.

alginate solution and COCs were added to it. A 20 μL micropipette was used to release COCs in IVM medium in the inner part of the syringe, along the extrusion direction, in order to obtain and facilitate the automatized encapsulation of one COC/microbead (Fig 2A). As described above, COC-microbeads were collected approximately 2 minutes after the initiation of the formation process and kept immersed in the $CaCl_2$ solution to allow homogeneous crosslink formation. The microbeads were then observed under an optical stereomicroscope and only those containing one COC were selected and used for IVM. The "encapsulation efficiency" was measured before IVM as "encapsulation rate", the proportion of COCs loaded in the Sphyga syringe which came out as included in microbeads in a 1/1 proportion. Those COCs which resulted non included in microbeads or included in a more than 1/1 proportion (in groups of 2 or more COCs) were excluded.

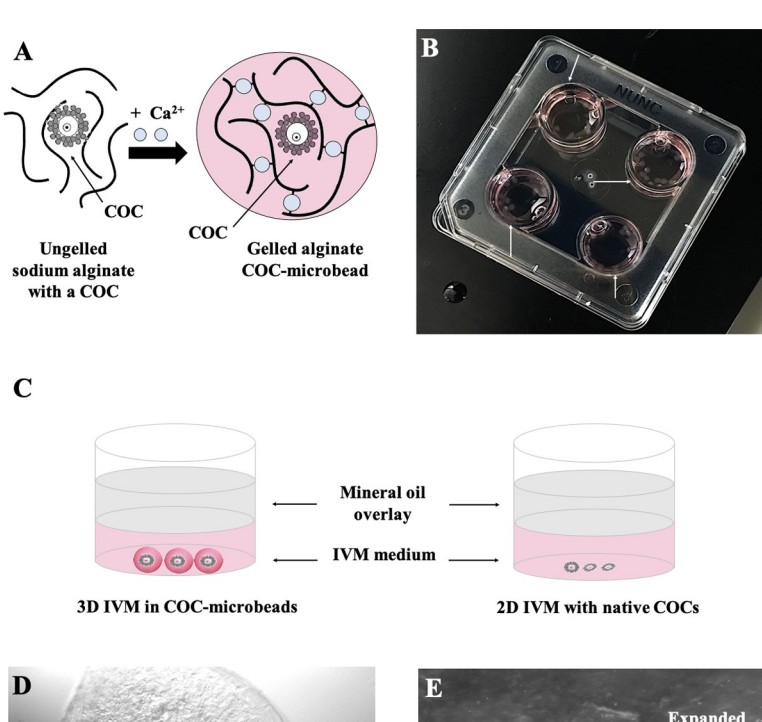

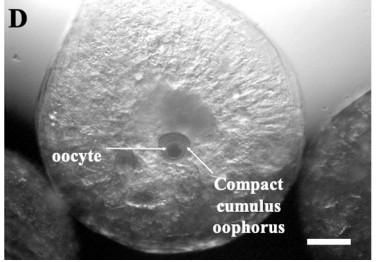

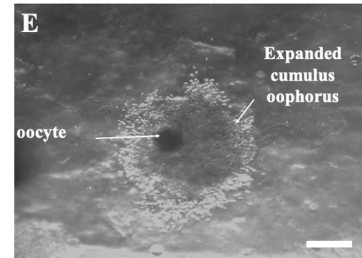

**Fig 2. 3D IVM set up and cumulus expansion assessment.** (A) Calcium-induced alginate gelification in presence of a COC to produce a COC-microbead. (B) COC-microbeads placed in 4-well culture dishes for IVM (white arrows). (C) Diagram of oocytes cultured under 3D and 2D IVM conditions. (D, E) Representative images showing a 1% alginate COC-microbead as observed before (D, with compact cumulus oophorus, scale bar represents 400 μm) and after 24 hours 3D IVM culture (E, with expanded cumulus oophorus, scale bar represents 300 μm).

## Three-dimensional (3D) IVM

Selected COC-microbeads were rinsed with fresh IVM medium and placed in the culture plates (mean n. of microbeads/well = 20–25; Fig 2B and 2C). COCs not subjected to encapsulation were cultured in the same dishes for use as conventional 2D controls (mean number of COCs/well = 20–25; Fig 2C) [54]. For both 3D and 2D IVM systems, the overall process, from COC selection/encapsulation until place the culture plate in the incubator took about 2 hours. *In vitro* maturation was performed for 24 hours at 38.5˚C under 5% $CO_2$ in air. After 24 hours IVM, cumulus expansion was analyzed meanwhile the COC was in the microbead under stereoscopic microscopy (Nikon SMZ1500, Nikon Instruments, Amsterdam, The Netherlands). Then, sodium alginate was removed by calcium chelation with 2% w/v sodium citrate in IVM medium for 5 minutes at 38.5˚C under 5% $CO_2$ in air. COCs were recovered and cumulus expansion was checked again. Indeed, COCs showing cumuli with continuous edges, consisting of cells in close contact each other, were classified as compact (Fig 2D), whereas cumuli showing discontinuous edges following cell detachment and production of a viscous extracellular matrix were classified as expanded (Fig 2E). The percentage of expanded/total COCs was reported. COCs underwent cumulus cell removal by incubation in TCM-199 with 20% FCS

containing 80 IU hyaluronidase/mL and aspiration in and out of finely drawn glass pipettes. Denuded oocytes were evaluated for meiotic stage and matured ones were used to assess bioenergetic/oxidative status assessment or gene expression.

## Staining for mitochondria and ROS

Oocytes were stained for mitochondria and reactive oxygen species (ROS) following previously reported procedures [54]. Briefly, oocytes were incubated for 30 minutes in IVM medium with 280 nM MitoTracker Orange CMTM Ros (Molecular Probes) at 38.5˚C under 5% $CO_2$ followed by incubation for 15 minutes in the same medium with 10 μM 2',7'- dichlorodihydrofluorescein diacetate ($H_2$DCF-DA) at 38.5˚C under 5% $CO_2$ in order to detect dichlorofluorescein (DCF)-stained intracellular ROS. After staining, oocytes were fixed overnight at 4˚C with 2% (v/v) paraformaldehyde solution in PBS until processed for oocyte nuclear chromatin evaluation as described below.

## Oocyte nuclear chromatin evaluation

Fixed oocytes were stained with 2.5 μg/mL Hoechst 33258 in 3:1 (v/v) glycerol/PBS and mounted on microscope slides with coverslips, sealed with nail polish and kept at 4˚C in the dark until observation. Slides were examined under an epifluorescence microscope (Nikon Eclipse 600; ×400 magnification) equipped with a B-2A (346 nm excitation/460 nm emission) filter. Oocytes were evaluated in relation to their meiotic stage and classified as germinal vesicle (GV), metaphase to telophase I (MI to TI), MII with the 1st polar body extruded, or as abnormal [54]. Oocytes showing either multipolar meiotic spindle or irregular chromatin clumps or absence of chromatin were considered as abnormal [60]. Oocytes with the first polar body extruded and one pronucleus were classified as parthenogenetically activated [61].

## Assessment of mitochondrial distribution pattern

Matured oocytes were examined at 600x magnification in oil immersion with a Nikon C1/ TE2000-U confocal laser scanning microscope (CLSM); 25 optical series were made per oocyte with a step size of 0.45 μm. The mitochondrial distribution pattern was evaluated following previously reported criteria [54]. Thus, homogeneous distribution of small mitochondria aggregates throughout the cytoplasm indicates weak bioenergetic status whereas perinuclear (with mitochondria around meiotic spindle) associated with subplasmalemmal (in the cortical region) distribution was considered as healthy bioenergetic status (P/S; [54]).

## Fluorescence intensity quantification

In each individual MII oocyte, MitoTracker Orange CMTM Ros and DCF fluorescence intensities were measured at equatorial plane, with the EZ-C1 Gold Version 3.70 image analysis software for Nikon C1 confocal microscope. A circle area was drawn to measure the area including cell cytoplasm and fluorescence intensity encountered within the scan area was recorded and plotted against a conventional pixel unit scale (0–255). Sample signals were expressed as percentage of the control sample. [54, 62].

## Mitochondria-ROS colocalization analysis

Colocalization analysis of mitochondria and ROS was performed with the EZ-C1 Gold Version 3.70 software. Degree of colocalization was reported as correlation coefficient quantifying the overlap degree between MitoTraker Orange CMTM Ros and DCF fluorescence signals [54, 62].

## Quantitative RT-PCR

Additional independent replicates of 3D versus 2D IVM were performed (30–35 COC-microbeads/well for 3D IVM and 30–35 COCs/well for 2D IVM). After 24 hours IVM, cumulus cells were removed. Denuded oocytes were evaluated under a stereomicroscope (Olympus SZH10, Italy) and those showing the first polar body extruded were selected. Total RNA was extracted with an RNAeasy Mini Kit (Qiagen, Valencia CA, USA) as per manufacturer's instructions. For each replicate of each experimental group, RNA was extracted from pools of 20–26 matured oocytes/pool. Total RNA (18 μL from each sample) was kept at 70°C for 5 minutes and then chilled on ice. Ten μL of RNA were used for reverse transcription-polymerase chain reaction (RT-PCR). Reverse transcription was performed in a volume of 20 μL, consisting of 10 μL of samples and 10 μL of a mastermix containing 4 μL 5 × 1$^{st}$ strand buffer, 0.4 μL random primers (0.09 IU/mL), 0.2 μL RNAse (40 IU/mL), 0.75 μL Superscript III (200 IU/mL) (Invitrogen, Groningen, The Netherlands), 2 μL dithiothreitol (0.1 M), 1 μL dNTP mix (10 mM) and 1.65 μL $H_2O$, and incubated at 50°C for 1 hour. As a negative control, reverse transcriptase was replaced by $H_2O$ (-RT blanks). Samples were subsequently kept at 80°C for 15 minutes and stored at −20°C. The PCR mixture contained 1 μL cDNA, 0.125 μL forward and reverse primers (0.5 μM each; Isogen, Maarssen, The Netherlands), 12.25 μL iQ SYBR Green supermix (Bio-Rad Laboratories, Hercules, CA, USA) and 11.25 μL $H_2O$ for a total volume of 25 μL. The list of analysed genes is reported in Table 1. The oligonucleotide primers were designed using Primer-BLAST (http://www.ncbi.nlm.nih.gov/tools/primer-blast/) using default criterion of the software with amplified products ranging from 100 to 400 bp. Primer amplification efficiency was determined from standard curves generated by serial dilutions of cDNA (5-fold each) for each gene in triplicate. After an initial denaturation step at 95°C for 3 minutes, 40 cycles were carried out each consisting of 95°C for 15s, the primer specific annealing temperature (Table 1) for 10s, and 72°C for 20s. Quantitative RT-PCR was performed in

**Table 1. List of analyzed genes.**

| Gene symbol | Gene description | GenBank accession n. | Primers | Annealing Temperature (°C) | Size (bps) |
|---|---|---|---|---|---|
| TFAM | Mitochondrial Transcription Factor A | XM_027962472 | F: 5' CTCGCGGCTTTAGGGCG 3'<br>R: 5' CCACTCAAGCTGGATGAAAACC 3' | 61 | 235 |
| ATP6 | Mitochondrially encoded ATP synthase F$_O$ subunit 6 | AF010406 | F: 5' CTACCACAAGGGACACCCAC 3'<br>R: 5' ATTAGTGCAAGGGTGGCTCC 3' | 58 | 161 |
| ATP8 | Mitochondrially encoded ATP synthase F$_O$ subunit 8 | AF010406 | F: 5' GCCACAACTAGACACATCAACG 3'<br>R: 5' AGGGGTAATGAAAGAGGCAAA 3' | 61 | 195 |
| KHDC3 | KH domain-containing protein 3 | XM_004011872 | F: 5' CAGACCCTGCTTCACGTTCA 3'<br>R: 5' CTTCTCAGAGCTTCGCGCC 3' | 60 | 150 |
| NLRP5 | NLR family, pyrin domain containing Protein 5 | HM037368.1 | F: 5' CAGCCTCCAGGAGTTCTTTG 3'<br>R: 5' GACAGCCTAGGAGGGTTTCC 3' | 59 | 212 |
| OOEP 1 | Oocyte Expressed Protein 1 | KF218578 | F: 5' ATCCGCTGGTGTTCTTCCTG 3'<br>R: 5' GAACACGGTGACTTCGACCA 3' | 60 | 149 |
| OOEP 2 | Oocyte Expressed Protein 2 | KF741040 | F: 5' TCCCCAAACTCCTTGCAGTG 3'<br>R: 5' CGGCAGGTAGGTGTCTGAAT 3' | 60 | 114 |
| TLE6 | Transducin-Like Enhancer of Split 6 | XM_004009373 | F: 5' TACCTGCGCACCTGCCTGCT 3'<br>R: 5' ATTGGTGAAGCCAGCAAAAG 3' | 58 | 195 |
| GAPDH | Glyceraldehyde-3-phosphate dehydrogenase | NM_001190390 | F: 5' GGTTGTCTCCTGCGACTTCA 3'<br>R: 5' CAGGGCCTTGAGGATGGAAA 3' | 60 | 301 |

duplicate on two replicates of cDNA and singular on the −RT blanks. For each target gene, all the samples were quantified simultaneously in one run in a 96-well plate using a real-time PCR detection system (MyiQ Single-color, Real-Time Detection System; Bio-Rad). Melting curves were plotted to verify single product amplification. Standard curves made on cDNA dilutions were used to calculate the relative starting quantity of each experimental sample. The expression levels Actin Beta (*ACTB*) and glyceraldehyde-3-phosphate dehydrogenase (*GAPDH*) were analyzed for normalization, but the expression of *ACTB* demonstrated to be rather dynamic and not suitable for normalization. Therefore, only *GAPDH* was used for normalization of the expression levels of the target genes performed by using the ratio of the relative starting quantity of the target gene with the reference gene.

## Statistical analysis

Microbead morphology data were compared by the Student's t test. For evaluation of microbead stability, mean values (mean±standard deviation) were compared among groups by one-way ANOVA (Tukey's multiple comparisons test). Encapsulation and cumulus expansion rate, as well as the proportions of oocytes showing different chromatin configurations and mitochondrial distribution pattern were compared between groups by Chi-square test. For mitochondria and ROS quantification analysis, raw values of fluorescence intensities and correlation coefficients were compared between groups by unpaired Student's t test. Mean values (mean ± standard deviation) of fluorescence intensities and mitochondria-ROS colocalization, indicated as correlation coefficient, were expressed as percentage of the signal of the control sample. Gene expression data were compared by the Student's t test. Differences with $P < 0.05$ were considered to be statistically significant.

## Results

### Microbead fabrication and characterization

The first objective was to obtain microbeads of adequate size and shape to contain a sheep COC which, like the human COC, measures around 300 μm and reaches around 500–600 μm in diameter after cumulus expansion. For this purpose, needles of different inner diameters and two alginate concentrations, 1% and 2%, were used. The microbeads were evaluated by acquiring bright-field images immediately after fabrication. The needle with a 508 μM diameter (21 G) was the working parameter which gave rise to reproducible microbeads of the desired size (1000 μm in radius) and shape, suitable for inclusion of ovine COCs (Table 2). In these conditions, reproducible and homogeneous spherical beads with an aspect ratio = 1.21 ± 0.10 (i.e. nearly circular) were obtained. As observed immediately after fabrication, no significant size and shape differences were observed in the beads obtained with 1% versus 2% alginate (Table 2). Then, the suitability of the two examined alginate concentrations was evaluated in terms of COC encapsulation and cumulus expansion efficiency. COCs in 2 independent replicates were used (n = 100 COCs/replicate). Just after fabrication, COC encapsulation rate in 1% alginate was significantly higher compared with 2% alginate (P<0.0001; Table 3). In both alginate concentrations this value ranged between 70 and 100%. After IVM, cumulus expansion rate was also significantly higher in 1% alginate microbeads compared with 2% alginate (P<0.0001; Table 3). These results led us to prefer the use of 1% alginate for subsequent studies. Therefore, microbead stability and oocyte maturation rate were analyzed in 1% alginate and by using the 508 μm needle. 3D microbeads produced with 1% alginate and the 508 μm needle remained stable in shape and size when incubated over 24 hours at 38.5°C under 5% $CO_2$ in air (Table 4). No broken beads were observed at any time.

**Table 2. Microbead morphology immediately after fabrication as related to needle size and alginate concentration.**

| Needle internal diameter (µm) | Alginate concentration (%) | Radius (µm) | Aspect ratio (*) |
|---|---|---|---|
| 610 | 1 | 1148.20 ± 151.48 | 1.71 ± 0.33 |
| | 2 | 1179.53 ± 204.95 | 1.75 ± 0.25 |
| 508 | 1 | 1070.39 ± 88.23 | 1.21 ± 0.10 |
| | 2 | 1099.69 ± 112.46 | 1.33 ± 0.18 |
| 406 | 1 | 900.90 ± 110.18 | 1.44 ± 0.23 |
| | 2 | 925.49 ± 107.83 | 1.58 ± 0.21 |
| 330 | 1 | 859.33 ± 17.30 | 1.39 ± 0.18 |
| | 2 | 882.77 ± 47.31 | 1.42 ± 0.26 |
| 152 | 1 | 807.44 ± 47.49 | 1.28 ± 0.16 |
| | 2 | 759.44 ± 58.29 | 1.40 ± 0.29 |
| 102 | 1 | 723.02 ± 61.81 | 1.30 ± 0.14 |
| | 2 | 703.85 ± 63.24 | 1.43 ± 0.20 |

(*) Aspect ratio defined according to Russ [51]. For each condition, data are mean ± standard deviation of n = 50 microbead measures. Student's t test: for each needle size, comparisons 1% versus 2% alginate were not significant.

### 3D IVM improves oocyte nuclear maturation

In the IVM study, 349 COCs were cultured under 3D IVM within 1% alginate engineered microbeads obtained with a 508 µm needle (n = 196) versus 2D IVM (n = 153) in 6–9 independent replicates. After culture under 3D IVM, the percentage of oocytes which reached the MII stage was significantly higher than that obtained after culture in the conventional control 2D system (P<0.05, Table 5). Correspondingly, significant reduction of the percentage of oocytes showing abnormal chromatin configuration, compared with the 2D controls, was observed (P<0.05, Table 5). In 2% alginate microbeads, oocyte maturation rate was significantly lower (P<0.05) whereas the activation rate was significantly higher in respect with 2D controls (P<0.001; S1 File).

### 3D IVM improves oocyte bioenergetic/oxidative status

To determine the effects of 3D IVM on oocyte bioenergetic/redox status, matured oocytes derived from 3D (n = 29) versus 2D IVM (n = 14) underwent mitochondria and ROS staining followed by CLSM analysis. No effect on the distribution of mitochondria was observed after 3D IVM. Indeed, the percentages of oocytes showing heterogeneous perinuclear and subcortical mitochondrial distribution, indicating cytoplasm maturity and competence, were similar in the two groups (24/29, 83% versus 11/14, 79%; for 3D and 2D respectively; Chi-square test, not significant). However, oocyte mitochondrial membrane potential and intracellular ROS levels significantly increased after 3D IVM, as assessed by increased MitoTracker Orange CMTM Ros and DCF fluorescence intensities, respectively (Fig 3A, red and green bars). Mitochondrial/ROS colocalization did not vary between the two IVM procedures (Fig 3A, yellow

**Table 3. Effects of alginate concentration on encapsulation and expansion rate of adult sheep COCs in engineered microbeads.**

| Alginate concentration (%) | n. of analyzed microbeads | Encapsulation rate n. (%) | Cumulus expansion rate n. (%) |
|---|---|---|---|
| 1 | 100 | 98 (98) [a] | 85/98 (87) [a] |
| 2 | 100 | 71 (71) [b] | 36/71 (51) [b] |

Chi square test: within each column, different superscripts indicate statistically significant differences: a, b = P<0.0001.

**Table 4. Microbead stability in IVM culture expressed as size (radius) and the shape descriptor aspect ratio (\*), characterized as function of 4-hours intervals.**

| Time (h) | Radius (µm) | Aspect ratio |
|----------|-------------|--------------|
| 0 | 1058.51 ± 117.01 | 1.17 ± 0.10 |
| 4 | 985.44 ± 106.23 | 1.18 ± 0.10 |
| 8 | 1019.36 ± 87.25 | 1.20 ± 0.14 |
| 12 | 1038.81 ± 95.18 | 1.22 ± 0.12 |
| 16 | 1020.16 ± 85.02 | 1.22 ± 0.13 |
| 20 | 1012.55 ± 98.38 | 1.22 ± 0.12 |
| 24 | 1053.99 ± 72.15 | 1.18 ± 0.10 |

(\*) Aspect ratio defined according to Russ [51]. For each condition, data are mean ± standard deviation of n = 50 microbead measures. Microbeads fabricated with 508 µm needle and 1% alginate. One-way ANOVA with post-hoc Tukey's multiple comparisons test: not significant.

bars). Fig 3B shows representative photomicrographs of an oocyte cultured in 3D IVM (lane 1) and a control oocyte (lane 2). Both oocytes show P/S mitochondrial distribution and the oocyte matured in the 3D IVM system shows increased mitochondria- and ROS-related fluorescence intensities (columns C and D).

## 3D IVM modulates oocyte gene expression

To investigate the effect of 3D IVM culture on oocyte molecular maturation the relative abundance of a panel of genes expressed after oocyte maturation was analyzed (Table 1). After additional trials (n = 3) of 3D versus 2D IVM, denuded oocytes showing the first polar body extruded (77/96, 80% and 72/106, 68%; for 3D and 2D IVM respectively; P<0.05) were pooled in groups of 20–26 oocytes/group, processed and examined. The expression of genes related to oocyte mitochondrial activity (*TFAM*, *ATP*6 and *ATP*8) was significantly upregulated in oocytes matured in the 3D IVM system compared to controls (Fig 4A). The expression of genes coding for SCMC proteins (*KHDC3*, *NLRP5*, *OOEP* and *TLE6*) associated with oocyte developmental competence, was also analyzed. The expression of *KHDC3* and *NLRP5* was significantly downregulated in oocytes matured in the 3D IVM system compared to control oocytes (Fig 4B). The relative abundance of *OOEP1*, *OOEP2* and *TLE6* transcripts was similar between groups (P-values: 0.203, 0.124 and 0.872 for *OOEP1*, *OOEP2* and *TLE6*, respectively; Fig 4C).

## Discussion

An efficient 3D system, capable of improving IVM culture conditions, has to ensure COC integrity as this is a basic aspect of oocyte trophic potential. To the best of our knowledge, this is the first study proposing an automated one-step bioprinting method for preparation of

**Table 5. Effects of 3D IVM culture in 1% alginate engineered microbeads on the maturation rate of adult sheep oocytes.**

| IVM system | n. of analyzed COCs | Nuclear chromatin configurations n. (%) | | | |
|------------|---------------------|------|-----------|-----|----------|
| | | GV | MI to TI | MII | Abnormal |
| **2D** | 153 | 26 (16.9) | 21 (13.7) | 87 (56.9) [a] | 19 (12.4) [a] |
| **3D** | 196 | 29 (14.8) | 24 (12.2) | 133 (67.9) [b] | 10 (5.1) [b] |

GV, Germinal vesicle; M, metaphase; T, telophase. Chi square test: within each column, different superscripts indicate statistically significant differences: a, b = P<0.05.

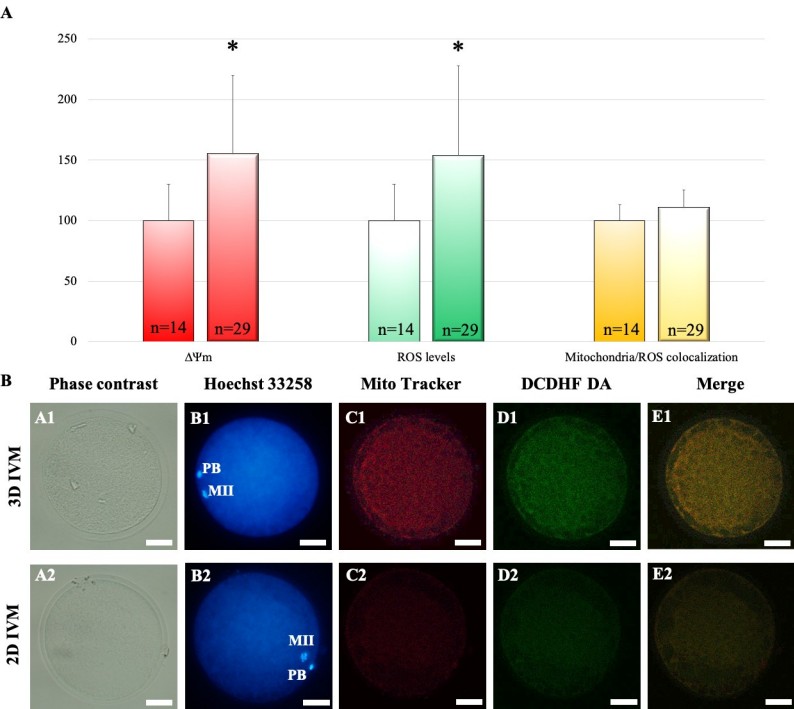

**Fig 3. Effects of 3D IVM on bioenergetic/oxidative status.** (A) Quantification data of bioenergetic/oxidative parameters of MII oocytes cultured in 2D (n = 14 MII oocytes) versus 3D IVM (n = 29 MII oocytes). Mitochondrial activity and intracellular ROS levels are expressed as percentage of the signal value of control samples (oocytes cultured in 2D IVM). Mitochondrial/ROS colocalization is expressed as percentage of the correlation coefficient value of control samples. Numbers of analyzed matured oocytes are indicated at the bottom of each graph bar. Student's t test: *$P < 0.05$. (B) Photomicrographs showing representative images of oocyte after 3D (lane 1) and 2D IVM (lane 2). Corresponding phase-contrast images showing cell morphology (column A), epifluorescence images showing nuclear chromatin configuration (column B: Hoechst 33258) and confocal images showing perinuclear and subplasmalemmal (P/S) mitochondrial distribution pattern and activity (column C: MitoTracker Orange CMTM Ros), intracellular ROS localization and levels (column D: DCDHF DA) and mitochondria/ROS colocalization (column E: Merge). Confocal images were taken at the oocyte equatorial plane. Scale bars represent 40 μm.

COC-microbeads and assessing its validity on oocyte nuclear and cytoplasmic maturation. Indeed, most of the studies published to date described two-step or manual procedures [32–36]. Dorati et al. [15] described alginate microbead preparation by the diffusion setting gelation technique in which, using a mouth-controlled hand-pulled glass Pasteur micro-pipette, single COCs were transferred into the alginate solution in culture medium and dropped into single wells of a 96-well plate containing the cross-linking agent. Morselli et al. [32] developed a two-step technique in which barium alginate microcapsules were produced and subsequently injected feline and canine COCs into the inner core of the microcapsules [32–34, 36]. Faustini et al. [35] described a protocol for bioencapsulation of COCs and granulosa cells in which a suspension of granulosa cells in cross-linking solution was dropped in an alginate solution. The COCs were then injected into the capsules and immersed in culture media in wells. As far as we know, the present study is the first reporting a method in which COCs are encapsulated meanwhile the microbead is formed, thus reducing the number of passages and manipulations to which COCs are subjected. This method is highly reproducible and capable of controlling cumulus size and integrity. It is thus applicable for clinical and research purposes and for regulatory testing.

Alginate hydrogel was the biocompatible matrix of choice as it has been reported for *in vitro* culture of pre-antral follicles of several animal species. In recent studies, the use of this

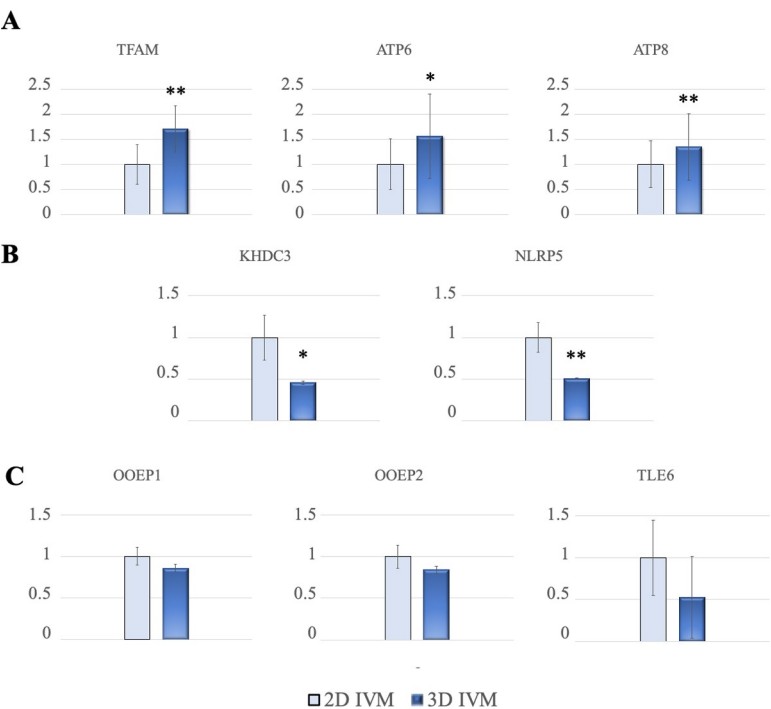

**Fig 4. Effects of 3D IVM on oocyte transcript abundance for genes related with mitochondria function and maternal effect (developmental potential).** (A) Relative expression of *TFAM*, *ATP6*, *ATP8*. (B) Relative expression of *KHDC3* and *NLRP5*. (C) Relative expression of *OOEP1*, *OOEP2* and *TLE6*. Comparisons between data obtained from ovine mature (MII) oocytes derived from 3D versus 2D IVM. Data are normalized by using the ratio of the relative starting quantity of the target gene with the reference gene (*GAPDH*). For each gene, three replicates were performed. Student's t test: *P<0.05; **P<0.01.

biomaterial has also been reported for oocyte IVM [15, 32–36]. The clinical interest in IVM is constantly urged by the World Health Organization since infertility affects about 15% of couples [15]. Moreover, women who undergo cytotoxic treatments for cancer therapy require the use of cryopreserved oocytes and could potentially benefit from more efficient IVM protocols. In addition, IVM is the elective procedure for the *in vitro* production of embryos for the farm animal production industry [9]. Our data contribute to the validation of alginate hydrogel in ARTs, as successful oocyte culture was performed in which oocytes reached nuclear and cytoplasmic maturation.

The Sphyga bioprinter allowed the automated generation of high numbers of microbeads with homogeneous size and shape [38]. Thus, it was also possible to establish a fixed number of COC-microbeads per well in culture dishes. The analysis of the effects of needle diameter on microbead size and morphology was used to obtain microbeads with similar size and spherical shape to try to ensure equal distance from the COC surface to the outer environment and to enhance gas exchange, nutrient diffusion and cellular waste removal. Indeed, immature COCs from adult ewes are 200–300 μm in diameter. After IVM, COCs reach 500–600 μm due to cumulus expansion. For this reason, a microbead of 1000 μm in radius can suitably house an expanded COC. The possibility of controlling microbead dimensions is advantageous for potential applications to COCs of different animal species, as COCs of different species vary in size due to differences in cumulus cell layer number.

The subsequent part of the study concerned the effects of the 3D IVM system on oocyte nuclear and cytoplasmic maturation. After IVM, it was evident that the 3D system allowed full

cumulus expansion, improved oocyte nuclear maturation rates and reduced the incidence of meiosis abnormalities. In addition, mitochondria activity and ROS generation ability were significantly increased after 3D IVM. These data were associated with unaltered colocalization of intracellular ROS with active mitochondria, denoting well-balanced oocyte redox status [54, 62, 63]. The CLSM-based multiparametric method used in the present study allowed to observe qualitative (mitochondria distribution pattern) and quantitative (mitochondrial activity and ROS generation ability) aspects of oocyte bioenergetic/oxidative status and to correlate them by assessing mitochondria—ROS colocalization, as biomarker of healthy, non oxidized, cytoplasmic condition. Mitochondria are fundamental organelles for oocytes since they regulate a number of cellular functions including ATP generation, calcium homeostasis, regulation of cytoplasmic redox state and signal transduction. So, optimal mitochondria function is required for oocyte maturation. Indeed, oocytes with defect in mitochondria activity such as depleted mitochondrial membrane potential, inhibited oxidative phosphorylation and consequent reduced ATP production experience problems with maturation and subsequent embryo development [64]. On the contrary, higher ATP levels in oocyte correlate with better embryo development and implantation rate [64]. As mitochondrial dysfunction negatively affects reproduction, improvement of mitochondrial function could potentially lead to improved fertility.

Increased levels of *TFAM*, *ATP6* and *ATP8* gene expression were found which can be considered as predictive biomarkers of oocyte mitochondrial function [41]. During their maturation, oocytes store a large amount of mitochondria-related RNAs, involved in the translation of proteins of the respiratory chain, useful for production of ATP, necessary to support the energy demands for the various mitochondrial activities, including their intracellular motility, maintenance of homeostasis, and regulation of cell survival processes during fertilization, early embryonic development and genome activation [40]. Indeed, decreased *ATP6* and *TFAM* expression during oocyte meiosis inhibits the capacity for oxidative phosphorylation and influence ATP generation, leading to chromosomal nondisjunctional error and aneuploidy since the required energy for spindle assembly has not been supplied [65, 66]. The effects of 3D IVM on oocyte developmental potential were analyzed through the subcortical maternal complex (SCMC) gene expression analysis. Expression of SCMC transcripts has been reported in oocytes and *in vitro* produced early embryos in sheep [47]. It has been demonstrated that mRNA expression is maximal in fully grown oocytes and abruptly decreases during oocyte maturation and early development as part of the overall program of MEG transcript degradation and embryo genome activation [47]. Our findings revealed significant reduction of *NLRP5* and *KHDC3* expression in oocytes matured in the 3D compared with the 2D system, suggesting improved oocyte developmental competence. On the other hand, the expression of *TLE6* remained stable in our study. This result is also in agreement with that by Bebbere et al. [47] of unchanged expression between low and high competence oocytes and embryos. TLE6 is important for the formation of the cytoplasmic F-actin meshwork and its absence leads to an unproper symmetric cell division and cleavage-stage embryonic death [50]. The unchanged expression level of this gene in our study could be interpreted considering the necessary persistence, throughout embryo development, of cytoplasmic structures driving symmetric blastomere divisions. In addition, our data showed unvaried expression of the OOEP transcript variants, *OOEP1* and *OOEP2*. Our data on effects of 3D IVM on gene expression are in agreement with those by Morselli et al. [34] who reported down-regulation of the growth differentiation factor-9 (*GDF-9*), another gene involved in oocyte developmental competence, in canine oocytes after 3D IVM.

Overall, these results indicate that the 3D IVM culture improved nuclear maturation and the expression of some mitochondrial function and maternal effect genes. These results could

be explained hypothesizing that the maintenance of the cumulus-oocyte 3D structure, during IVM culture, could have contributed to increase of the quantitative and/or qualitative relationship between regulatory factors and their receptor sites, thus improving the efficacy of signaling pathways of meiotic maturation. Further studied are necessary to confirm this hypothesis.

In conclusion, this study reports an automated one-step bioprinting-based technique for production of COC-microbeads of reproducible size and shape and demonstrates that this 3D IVM method improves oocyte nuclear maturation and cytoplasmic parameters, biomarkers of oocyte bioenergetic and developmental potential.

## Supporting information

**S1 File. Effects of 3D IVM culture in alginate engineered microbeads on the maturation rate of adult sheep oocytes.**
(DOCX)

## Author Contributions

**Conceptualization:** Arti Ahluwalia, Maria Elena Dell'Aquila.

**Data curation:** Antonella Mastrorocco.

**Formal analysis:** Antonella Mastrorocco, Maria Elena Dell'Aquila.

**Funding acquisition:** Elena Ciani, Maria Elena Dell'Aquila.

**Investigation:** Antonella Mastrorocco, Ludovica Cacopardo, Diana Fanelli.

**Methodology:** Antonella Mastrorocco, Ludovica Cacopardo.

**Project administration:** Elena Ciani, Maria Elena Dell'Aquila.

**Resources:** Francesco Camillo, Elena Ciani, Bernard A. J. Roelen, Arti Ahluwalia, Maria Elena Dell'Aquila.

**Software:** Antonella Mastrorocco, Ludovica Cacopardo, Arti Ahluwalia.

**Supervision:** Nicola Antonio Martino, Elena Ciani, Bernard A. J. Roelen, Arti Ahluwalia, Maria Elena Dell'Aquila.

**Validation:** Nicola Antonio Martino, Elena Ciani, Bernard A. J. Roelen, Arti Ahluwalia, Maria Elena Dell'Aquila.

**Visualization:** Antonella Mastrorocco, Nicola Antonio Martino, Arti Ahluwalia, Maria Elena Dell'Aquila.

**Writing – original draft:** Antonella Mastrorocco, Maria Elena Dell'Aquila.

**Writing – review & editing:** Antonella Mastrorocco, Ludovica Cacopardo, Nicola Antonio Martino, Diana Fanelli, Francesco Camillo, Elena Ciani, Bernard A. J. Roelen, Arti Ahluwalia, Maria Elena Dell'Aquila.

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
