## [Decision Letter · Decision Letter 0]

5 May 2020

PONE-D-20-09540

One-step automated  method for cumulus-oocyte complex microencapsulation

for 3D in vitro maturation

PLOS ONE

Dear Dr. Mastrorocco,

Thank you for submitting your manuscript to PLOS ONE. After careful consideration, we feel that it has merit but does not fully meet PLOS ONE’s publication criteria as it currently stands. Therefore, we invite you to submit a revised version of the manuscript that addresses the points raised during the review process.

Please, take in consideration all the reviewers' suggestions/corrections but especially those from reviewer #2 since he was the more stringent.

We would appreciate receiving your revised manuscript by Jun 19 2020 11:59PM. To enhance the reproducibility of your results, we recommend that if applicable you deposit your laboratory protocols in protocols.io, where a protocol can be assigned its own identifier (DOI) such that it can be cited independently in the future. For instructions see: http://journals.plos.org/plosone/s/submission-guidelines#loc-laboratory-protocols

We look forward to receiving your revised manuscript.

Kind regards,

Marcelo Fábio Gouveia Nogueira, Associate Professor, Ph. D.

Academic Editor

PLOS ONE

2. Please provide the name of the slaughterhouse.

https://iopscience.iop.org/article/10.1088/1758-5082/6/2/025009/meta

In your revision ensure you cite all your sources (including your own works), and quote or rephrase any duplicated text outside the methods section. Further consideration is dependent on these concerns being addressed.

Reviewers' comments:

Reviewer's Responses to Questions

**Comments to the Author**

1. Is the manuscript technically sound, and do the data support the conclusions?

Reviewer #1: Yes

Reviewer #2: Partly

Reviewer #3: Yes

2. Has the statistical analysis been performed appropriately and rigorously? 

Reviewer #1: I Don't Know

Reviewer #2: I Don't Know

Reviewer #3: Yes

3. Have the authors made all data underlying the findings in their manuscript fully available?

Reviewer #1: Yes

Reviewer #2: Yes

Reviewer #3: Yes

4. Is the manuscript presented in an intelligible fashion and written in standard English?

Reviewer #1: No

Reviewer #2: Yes

Reviewer #3: Yes

5. Review Comments to the Author

Reviewer #1: The article titled “One-step automated method for cumulus-oocyte complex microencapsulation for 3D in vitro maturation” the authors improved a 3D system already used in others cultures systems by turning it in an automated method that was proved to be stable during the 24 hours of in vitro maturation of ovine oocyte. Moreover, the authors demonstrated that a 3D in vitro maturation system can improve the ovine oocyte maturation. Compared to 2D usual system, the 3D system presented higher proportion of oocytes that completed the nuclear maturation and modulates the expression of genes previously described as markers of oocyte quality (i.e. TFAM, ATP6, ATP8, KHDC3, and NLRP5).

The work is potentially interesting with some level of novelty. However, the way the MS is written does not help me to follow the main idea of the authors. Therefore I do not recommend the MS for publication in its present form and I suggest a basic correction of the language (Introduction, Discussion) and a basic rewriting of the text (mainly Introduction and Discussion) in order to make to MS more reader friendly. I would like to give you some suggestions:

• Try to short the MS

• Check English

• Check repetition of abbreviations previously described (lines 152 and 180 for IVM, for example)

• Make the introduction more straight to your objective. It is too long, therefore in some point I got lost what is your point.

Other comments:

1. Line 144: Were ovaries from prepubertal lambs (under 6 months) also used in the experiments? It is not specified in the experiments (i.e., line 161), just in the “collection” section.

2. Figure 1 should be placed in the section “Alginate microbead fabrication” (Line 178: Figure 1A, 1B, 1C, 1D, 1E; Line 191: Figure 1F). It would help to comprehend the system.

3. How long was the process of automatized encapsulation of one COCs/microbead? For the experiment, you use 20/25 beads/well. How long the process takes to be ready (starting with COCs selection, COCs encapsulation, selection of microbeads containing one COC until place the plate in the incubator for 24h IVM)?

4. Figure 2E1 and 2F1 should be placed in the section “Three-dimensional (3D) in vitro maturation (IVM)” (Lines 225-227).

5. Line 222: how many microbeads containing COC were added in 400uL of IVM medium?

6. Line 226: how many COCs were added in 400uL of IVM medium for 2D culture system?

7. Line 229: What is the evaluation of “bioenergetic/oxidative status”?

8. For “Oocyte nuclear chromatin evaluation”, please specify the total number of oocytes analyzed per group (n=?), how many replicates, how many COCs per replicate per group.

9. For “Quantitative RT-PCR”, please specify the total number of oocytes analyzed per group (n=?), how many replicates, how many COCs per replicate per group.

10. Line 270: why only one reference gene was used (GAPDH)? How did you confirm it to be a reference gene?

11. How cumulus expansion rate was performed?

12. Why microbeads of 1000um were the desired size?

13. Add in the legend of Table 4 the information of needle size (508 uM) and alginate concentration (1%).

14. Line 331: Include the needle size used for the IVM study.

15. Lines 339-342: I would keep the results/table of maturation rates of COCs included in 2% alginate microbeads as supplementary results.

16. Line 409: Change “ithas” by “it has”.

17. Lines 418-419: Remove “developed to embryos at the preimplantation stages”, it wasn’t presented in this work.

18. Would this system applicable for in vitro embryo production?

Reviewer #2: In this manuscript, the authors described an optimization of the process of COC-­laden microbead by using a bioprinter (an automatic generator of hydrogel microspheres). The alginate microspheres produced were characterized in terms of size, shape and stability in culture conditions, using sheep as a large animal model. The results demonstrated high efficiency and reproducibility in the COC encapsulation, in which the cumulus integrity was preserved. After IVM, the 3D system supported oocyte meiotic maturation more efficiently than 2D control. In addition, 3D-matured oocytes exhibited increased expression of genes related to metabolic activation (TFAM, ATP6 and ATP8) and decreased expression of genes related to oocyte competence (KHDC3 and NLRP5) compared to 2D-matured oocytes. Based on the results, they concluded that the proposed method for the production of COC-laden microbeads has high reproducibility and efficiency and that oocyte competence is increased when maturation occurs in the 3D system.

Overall, the manuscript is well organized and the language is appropriate. However, in some parts of the Introduction and Methodology sessions, the text becomes wordy and repetitive.

The main question that has not been answered properly and, therefore, needs more arguments, is the statement and conclusion that the proposed method for 3D maturation improves the oocyte bioenergetic and development potential. Although different expression of genes related to metabolic activation and oocyte competence were observed when comparing oocytes matured in the traditional way (2D) and oocytes matured in 3D, no further evaluation was performed to prove that the difference in the level of the transcripts was actually reflected in different biological responses. For example, mitochondrial function and ATP production/consumption and embryonic development could have been assessed to confirm the increase in metabolic activation and oocyte competence, respectively. Thus, further data would be needed to support the conclusion.

Specific comments are:

Line 135: cytoplasmic competence was not investigated.

Line 138: For a better understanding of the development of the experiments, information related to the number of COC used, number of COC-laden microbeads cultured and number of structures evaluated in each analysis parameter should be presented in the methodology section, and not in the results. Likewise, the number of replicates in each experiment must also be informed.

Lines 144 and 161: author inform that ovaries from adult sheep (over 1 year) and prepubertal lambs (under 6 months) were recovered for the experiments, but only ovaries from adult cyclic ewes were used?

Line 164: minimum of how many layers?

Line 177: Have the authors previously tested the supply of oxygen and nutrients in the medium and the removal of waste or is it just an assumption? If it has been tested previously, quote the reference.

Line 229: Did the authors test the COC bioenergetic/oxidative status? These data were not presented.

Lines 278-279: microbead stability and microbead morphology were statistically compared between groups, however this analysis was not presented in the results (tables 1 and 4).

Line 315, Table 2: when was the analysis performed: immediately after formation or at the end of culture?

Line 320, Table 3: please inform if the evaluation was carried out at the end of the cultures.

Line 354, Table 5: even the highest observed M II rates are still low compared to the available literature. Do the authors have any explanation for this finding?

Line 363: the author can say that there was an increase in the expression of genes related to oocyte mitochondrial activity and embryonic development, but these parameters themselves were not evaluated. More data would be needed to make this statement.

Line 366: data differ from those reported in the table.

Line 374: please inform the P-value for the tendency.

Line 389: what does the author mean by oocyte competence? Meiotic competence was assessed in this study, but not competence for embryonic development. Please be more specific. Why has embryonic development not been evaluated?

Line 418: was embryo development evaluated?

Line 422-424: has a standardization study been carried out? If yes, present the data or quote the reference.

Line 426-427: these variables were not evaluated in the present study.

Line 438: oocyte meiotic competence?

Line 519: unnecessary information, as it is not related to the data presented in the study.

Reviewer #3: GENERAL COMMENTS

The authors devised a one-step automated system to encapsulate sheep cumulus-oocyte complexes (COCs) aiming for a 3D in vitro maturation (IVM) system, as opposed to the traditional 2D system. The 3D system would be a more physiological culture system as it would more closely preserve the natural COC-morphology and, therefore, would improve IVM outcomes. Authors have characterized the size and shape, and reproducibility, of microbeads produced with two different alginate concentrations, and also assessed the microbeads stability during a 24 h in vitro culture. IVM of COCs within the 3D microbeads in comparison to the usual 2D system (control) was evaluated in terms of oocyte nuclear maturation and expression of genes related with oocyte energy metabolism (mitochondrial function) and developmental competence (maternal effect genes related with oocyte competence). Authors determined the parameters for microbeads production which were repeatable and efficient, and stable in culture. Oocyte nuclear maturation increased in 3D culture and transcripts abundance were influenced by the culture system. The authors concluded that the 3D culture system using alginate microbeads was efficient and that it improved oocyte competence. The study is interesting, well designed, bringing new and promising results, and appears to have been appropriately conducted. Overall, the text is well written, organized and clear. There are few corrections, however, regarding some parts of the material and methods (MM) section in which some information is incomplete. Also, some information that should be in the Introduction is in the Discussion and some in the Results should be in MM. Some terms are not so adequate and conclusions are, in part, a little beyond what was presented in the Results, at least considering the way they were written. Therefore, the are suggestions to the text, which are addressed in the Specific comments.

Specific comments

Title OK

Absract

L29 – the term COC-laden for the microbeads seems a bit misleading, as laden gives the idea of something heavily loaded with something, and this is not the case as each microbead contains one COC, so the term laden would not be the best in this context. Probably individual COC encapsulated in microbeads or some other term would be better. If changed, do it throughout the text.

L41 – “… improves oocyte competence…” seems a strong conclusion in relation to the results, as few parameters were assessed (only nuclear maturation and expression of some genes); maybe improves some aspects/parameters of oocyte competence would be better. In the abstract, the conclusion put this way does not seem so much of a problem, as the size of the text may be limited, but this should be addressed, particularly in the final conclusions of the work.

Introduction

Some information regarding the genes chosen to be assessed as indicators of cytoplasmic competence should be given here. This information is only given in the Discussion.

Also, the sole expression of some mitochondrial function and maternal effect genes as indicators of cytoplasmic competence, as indicated in the text, does not seem the best description (L135-136); this assessment would be more a molecular competence, although these terms are sometimes used as synonyms; cytoplasmic competence would be best shown as organelle and/or cytoskeleton and/or specific proteins distribution in the cytoplasm and developmental competence would be best addressed using in vitro embryo development (either by IVF or parthenogenetic activation, for example). It would be better to be more specific and indicate that the assessment of the transcripts levels for selected maternal effect and mitochondrial function genes was chosen as an indicator of cytoplasmic or oocyte competence; or clearly indicate references that support that the chosen transcripts are directly related with and are markers of oocyte developmental competence.

As mentioned, the Introduction does not consider these factors – mitochondrial function and maternal effect genes – so it does not make clear the relation between cytoplasmic maturation and the assessments made in the experiments that were carried out.

Material and Methods

Part of the figures in Results include images of the setup of the system, and these parts of the figures could be placed in the MM. Sometimes only by the text it is difficult to understand how the microbeads were prepared, and mostly how the COCs individually were placed inside the microbeads. Therefore, the images referring to this, which are in Results, could be placed in MM to make it easier to understand the method.

L151 and L160 – As a suggestion, invert IVM medium and COC retrieval

L171-172 – are these notations correct – 5 divided by 20 and 1 divided by 100, and both between parenthesis?

L173 “…actuated…” would this be activated?

L176 – indicate here between parenthesis the range of desired dimensions

177 “…as well as…”

L185 – same as in L173

L205 – laden, could be changed to another term

L215 – here is when we finally get to know that there is one COC/microbead; it is not so easy to understand how, so referring to a figure here would be helpful.

L221-222 – how many microbeads with COCs inside in each well?

L226 – how many COCs/well?

L227 – time and temperature for chelation of calcium?

L229 – which bioenergetics/oxidative status? This was not measured, you did not analyse ROS levels or ATP levels or things like that, be more specific of what you´ve actually done

L229-232 No need to say “Those COCs destined for … and those destined for… underwent cumulus separation…” as all them were denuded. Just say that at the end of IVM COCs were denuded for analysis by incubation and so on, otherwise the sentence is repetitive.

L235 “… chromatin, oocytes…”

L245 – total RNA of how many oocytes / sample? What was the size of the oocyte pool? How many pools were used?

L257 – “The list of analysed genes is reported in Table 1” can be removed as the next sentence mentions the oligonucleotide primers and Table 1, so it is repetitive (or remove the Table 1 in parenthesis).

L276-277 the analysis of cumulus expansion rate is mentioned as analysed, but this analysis is not described; this description should be included in MM.

L279 morphology of the microbeads was also analysed, but no description of how this was done is provided; this should be included in MM.

Results

L291 – define “aspect ratio” and the meaning of being ≈1. Actually, a description of how the shape of the beads were analysed should be detailed in the MM

L294-298 – COC encapsulation rate and cumulus expansion rate – these should be explained how they were measured in MM; is there a difference between encapsulation efficiency, rate and levels? Specify what exactly each term means as this part is a bit confusing as it is.

L317 – as mentioned, the assessment of shape should be detailed in MM

L334 – cumulus expansion evaluation should be explained in MM

L338-339 – define what was considered abnormal chromatin configuration and activated oocytes in MM

L363 – “… improves oocyte energy and developmental potential” – actually, not quite this. Oocyte energy was not assessed, if so, there should be some measure of this parameter either by evaluating ATP production, mitochondrial activity or potential by specific staining, or other methods. You analysed the levels of transcripts for genes related with mitochondrial function, not directly oocyte energy. Otherwise, provide a reference that indicates that the expression of chosen genes can be directly linked to a cell´s energy levels. As there is nothing of this in the Introduction it is hard to find the direct connection authors are trying to make here. The same goes for developmental potential; no embryo production was evaluated. Again, the assessment was on transcript levels for maternal effect genes; could include reference indicating that expression of such genes are directly linked to developmental potential in vivo and/or in vitro; also here, no prior information on the genes is given in the Introduction. There was an effect on the transcript levels for genes related to mitochondrial function and maternal effect, which can affect developmental potential; this is an indirect assessment.

L366-367 – how were the 77 and 72 oocytes pooled in groups of 25 oocytes? this information should be in MM

L370 – what is SCMC? This is the first time in the manuscript this is mentioned and it only cited; then, it is only explained in the Discussion. This should be addressed in the Introduction or at least in the MM to justify the choice of genes to be studied (same for mitochondrial genes) and why they can be used as indicators of oocyte competence.

L374 – if there was a tendency what was the value of P?

L377 – it is not quite transcriptional activity; this is different from transcripts abundance or relative levels; probably best to use transcript abundance for genes related with mitochondrial function and maternal effects

Discussion

L389-390 – this part of the text mentions assessment of oocyte competence by cytogenetic and molecular analysis. Oocyte competence was not measured, but some parameters related to that, and cytogenetic analyses were not performed (the chromosomes were not studied, only chromatin configuration as indication of meiosis stage); nuclear maturation is not a cytogenetic analysis it is a cytological one. Attention with the use of some terms.

L418-419 – the way this part is written leads to the understanding that you have obtained the development of embryos; I understand that your results corroborate previous findings that showed such embryo development, but the phrasing is dubious, so it would be best to rephrase this part to make clear what exactly you are trying to inform.

L423 – this information is missing in MM

L438-443 – how does your MII rate compare to other 2D studies in sheep IVM? Is it similar to what was found in your 2D control system? And the rate of anomalies?

L448-459 – this entire part is more like a review explaining the genes that were studied; this could be (shorter even) in the Introduction and/or MM. Starting from the end of L459 is more like a discussion. Reorganize and shorten this part.

L471 – include “…could potentially improve oocyte mitochondria function” or provide reference that the increase in transcripts for these genes is a marker of enhanced mitochondrial function and make this information clear.

L472-512 this part mentions and then explains SCMC; this (also can be shorter) should be in the Introduction and/or MM (L370 just cites SCMC); the explanation on the genes goes up to L486; from here on in the text is more like a discussion contrasting your findings with their possible meaning for oocyte function. Reorganize and shorten this part.

L492- 493 you found accelerated degradation of maternal transcripts? Did you do kinetics of transcripts levels during IVM? What you found was that transcripts were decreased in 3D oocytes compared to 2D; rephrase this part to reflect what was observed.

L493 “…similar to the, as relationship…” it appears that there is some word or part missing after similar to the, is similar to the what?

L511 – “… potentially improved oocyte competence” or “… improved some parameters related with oocyte competence”, as assessment of competence was limited. It would be quite interesting in the future to evaluate effects on embryo development and quality and other functional parameters in oocytes and cumulus cells.

L516 - the conclusion oversteps a bit what was observed in the results, what was improved was nuclear maturation and the expression of some mitochondrial function and maternal effect genes; or make it clear throughout the text, with literature support, that these assessments (transcripts levels of chosen genes) can be used as reliable indicators of bioenergetic and developmental potential in oocytes. More studies are needed to further address the effects of the novel culture system on COC function, which seem to be, anyway, quite promising.

6. PLOS authors have the option to publish the peer review history of their article (what does this mean?). If published, this will include your full peer review and any attached files.

Reviewer #1: No

Reviewer #2: No

Reviewer #3: No

---

## [Author Response · Author response to Decision Letter 0]

15 Jun 2020

Dr. Antonella Mastrorocco

PhD in Functional and Applied Genomics and Proteomics

Department of Biosciences, Biotechnologies and Biopharmaceutics

University of Bari "Aldo Moro", Italy

Da: "PLOS ONE" <em@editorialmanager.com>

Oggetto: PLOS ONE Decision: Revision required [PONE-D-20-09540] - [EMID:72524dc1c2ad6652]

Data: 5 maggio 2020 17:18:52 CEST

A: "Antonella Mastrorocco" <antonella.mastrorocco@uniba.it>

Rispondi a: "PLOS ONE" <plosone@plos.org>

PONE-D-20-09540

One-step automated method for cumulus-oocyte complex microencapsulation for 3D in vitro maturation

PLOS ONE

Dear Dr. Mastrorocco,

Thank you for submitting your manuscript to PLOS ONE. After careful consideration, we feel that it has merit but does not fully meet PLOS ONE’s publication criteria as it currently stands. Therefore, we invite you to submit a revised version of the manuscript that addresses the points raised during the review process.

Please, take in consideration all the reviewers' suggestions/corrections but especially those from reviewer #2 since he was the more stringent.

We would appreciate receiving your revised manuscript by Jun 19 2020 11:59PM. To enhance the reproducibility of your results, we recommend that if applicable you deposit your laboratory protocols in protocols.io, where a protocol can be assigned its own identifier (DOI) such that it can be cited independently in the future. For instructions see: http://journals.plos.org/plosone/s/submission-guidelines#loc-laboratory-protocols

• A rebuttal letter that responds to each point raised by the academic editor and reviewer(s). This letter should be uploaded as separate file and labeled 'Response to Reviewers'.

• A marked-up copy of your manuscript that highlights changes made to the original version. This file should be uploaded as separate file and labeled 'Revised Manuscript with Track Changes'.

• An unmarked version of your revised paper without tracked changes. This file should be uploaded as separate file and labeled 'Manuscript'.

We look forward to receiving your revised manuscript.

Kind regards,

Marcelo Fábio Gouveia Nogueira, Associate Professor, Ph. D.

Academic Editor PLOS ONE

2. Please provide the name of the slaughterhouse.

The names of the slaughterhouses have been provided.

3. We noticed you have some minor occurrence of overlapping text with the following previous publication(s), which needs to be addressed: https://iopscience.iop.org/article/10.1088/1758-5082/6/2/025009/meta

We reduced as much as possible the overlapping text with the manuscript by Tirella et al., 2014.

In your revision ensure you cite all your sources (including your own works), and quote or rephrase any duplicated text outside the methods section. Further consideration is dependent on these concerns being addressed.

We cited all our sources, and quoted or rephrased duplicated texts outside the methods section, as requested.

Reviewers' comments:

Reviewer's Responses to Questions

Comments to the Author

1. Is the manuscript technically sound, and do the data support the conclusions?

Reviewer #1: Yes

Reviewer #2: Partly

Reviewer #3: Yes

2. Has the statistical analysis been performed appropriately and rigorously? 

Reviewer #1: I Don't Know

Reviewer #2: I Don't Know

Reviewer #3: Yes

3. Have the authors made all data underlying the findings in their manuscript fully available?

Reviewer #1: Yes

Reviewer #2: Yes

Reviewer #3: Yes 

4. Is the manuscript presented in an intelligible fashion and written in standard English?

Reviewer #1: No

Reviewer #2: Yes

Reviewer #3: Yes 

5. Review Comments to the Author

Reviewer #1: The article titled “One-step automated method for cumulus-oocyte complex microencapsulation for 3D in vitro maturation” the authors improved a 3D system already used in others cultures systems by turning it in an automated method that was proved to be stable during the 24 hours of in vitro maturation of ovine oocyte. Moreover, the authors demonstrated that a 3D in vitro maturation system can improve the ovine oocyte maturation. Compared to 2D usual system, the 3D system presented higher proportion of oocytes that completed the nuclear maturation and modulates the expression of genes previously described as markers of oocyte quality (i.e. TFAM, ATP6, ATP8, KHDC3, and NLRP5).

The work is potentially interesting with some level of novelty. However, the way the MS is written does not help me to follow the main idea of the authors. Therefore I do not recommend the MS for publication in its present form and I suggest a basic correction of the language (Introduction, Discussion) and a basic rewriting of the text (mainly Introduction and Discussion) in order to make to MS more reader friendly. I would like to give you some suggestions:

• Try to short the MS

• Check English

• Check repetition of abbreviations previously described (lines 152 and 180 for IVM, for example)

• Make the introduction more straight to your objective. It is too long, therefore in some point I got lost what is your point.

We thank the reviewer for her/his comments and suggestions. We modified the manuscript by shortening the Introduction to make it directly pertinent to our objective, as requested. In particular, we reduced the section of the bioprinting technology and moved the paragraph on expression of mitochondrial and maternal effect genes from Discussion to Introduction, to better clarify that these assessments were used as biomarkers of oocyte bioenergetic and developmental potential. 

We have therefore modified the manuscript so that the main objective is now clearer which is to demonstrate that the described automated microencapsulation method, in which encapsulation of a single COC/bead occurs meanwhile the bead is formed, allows producing microbeads of standardized shape and size, able to preserve cumulus integrity and stable for 24h. Moreover, the 3D IVM system improves nuclear maturation and some aspects of cytoplasmic maturation, as assessed by bioenergetic/oxidative status and expression of molecular biomarkers of mitochondrial activity and developmental potential. 

For bioenergetic potential, in the revised manuscript, also following the requests of the other reviewers, we added new confocal laser scanning microscopy analysis of mitochondria activity, reactive oxygen species (ROS) levels and mitochondria/ROS colocalization, performed on a group of 3D versus 2D matured oocytes. 

As biomarkers of oocyte developmental potential, we examined a panel of genes coding for the SubCortical Matermal Complex (SCMC), essential biomarker of oocyte meiotic spindle formation and positioning and for zygote progression beyond the first embryonic cell divisions (Li et al., 2008; Lu et al., 2018; Bebbere et al., 2014; 2016; Monk et al., 2017; Qin et al., 2019). 

The English language of the manuscript has been revised by a native speaker, expert in biomedical sciences.

Repetitions have been reduced and abbreviations have been checked, as requested

Other comments:

1. Line 144: Were ovaries from prepubertal lambs (under 6 months) also used in the experiments? It is not specified in the experiments (i.e., line 161), just in the “collection” section.

Ovaries from prepubertal lambs were not used in the present study. This indication has been removed.

2. Figure 1 should be placed in the section “Alginate microbead fabrication” (Line 178: Figure 1A, 1B, 1C, 1D, 1E; Line 191: Figure 1F). It would help to comprehend the system.

We thank the reviewer for this suggestion. Figure 1 has now been placed in the section “Alginate microbead fabrication”.

3. How long was the process of automatized encapsulation of one COCs/microbead? For the experiment, you use 20/25 beads/well. How long the process takes to be ready (starting with COCs selection, COCs encapsulation, selection of microbeads containing one COC until place the plate in the incubator for 24h IVM)?

The overall process, including IVM medium and IVM culture plate preparation, COC retrieval and selection, COC encapsulation, selection of microbeads containing one COC and placement in culture, takes around 2 hours. This time interval is the same as that used in the 2D system. Infact, in both systems, equilibration of IVM medium (1hour) and culture plate (30 min) in the incubator under controlled temperature (38.5°C) and gas atmosphere (5%CO2 in air) is important. This information has been added to the manuscript.

4. Figure 2E1 and 2F1 should be placed in the section “Three-dimensional (3D) in vitro maturation (IVM)” (Lines 225-227).

Figure 2 has been modified and included in the MM section.

5. Line 222: how many microbeads containing COC were added in 400uL of IVM medium?

A mean number of 20-25 COC-microbeads has been added in each well of a 4-well plate with 400µL of IVM medium. This detail has now been included in the methods section, as requested.

6. Line 226: how many COCs were added in 400uL of IVM medium for 2D culture system?

Each well contained 20-25 COCs. This detail has been addedd to the methods section. The same number of COCs per well was added for both culture systems in order to provide comparable conditions. 

7. Line 229: What is the evaluation of “bioenergetic/oxidative status”?

This specific sentence has been modified, as also suggested by other reviewers. Moreover, in the revised manuscript, we added confocal laser scanning microscopy data on the effects of 3D IVM on oocyte bioenergetic/oxidative status, expressed as mitochondria activity, intracellular ROS levels and mitochondria/ROS colocalization. This part has been implemented in Methods, Results and Discussion.

8. For “Oocyte nuclear chromatin evaluation”, please specify the total number of oocytes analyzed per group (n=?), how many replicates, how many COCs per replicate per group.

For oocyte nuclear chromatin evaluation, the total number of analyzed oocytes (now n=349) has been indicated in the Results section “3D IVM improves oocyte maturation rate”. This number has been now reduced because, according to the reviewer suggestion in point 15, the results/table of maturation rates in 2% alginate microbeads have been removed from the text and are now presented as a supplementary file (S1 File). The total number of analyzed oocytes per treatment group has been indicated in table 5. For each experiment and condition, at least six replicates were performed, in which a replicate is intended as a group of 20-25 COCs cultured for IVM in one well of a 4-well Nunc plate. The number of replicates (6 to 9) has now been included. 

9. For “Quantitative RT-PCR”, please specify the total number of oocytes analyzed per group (n=?), how many replicates, how many COCs per replicate per group.

For quantitative RT-PCR, three additional replicates per group were performed. In these experiments, for each replicate of each group, 30-35 COC were cultured under 3D versus 2D IVM and RNA samples were extracted from pools of 20-26 matured oocytes/group. This information has been added in the text in the methods section.

10. Line 270: why only one reference gene was used (GAPDH)? How did you confirm it to be a reference gene?

In order to quantify gene expression, qRT-PCR was used. The expression levels ACTB and GAPDH were originally chosen for normalization, but the expression of ACTB was rather dynamic and not suitable for normalization. Including both ACTB and GAPDH was actually less stable than the use of only GAPDH. Therefore, only GAPDH was used for normalization of the expression levels of the target genes.

11. How cumulus expansion rate was performed?

In this study, cumulus expansion analysis was performed while the COC was within the microbead and it was confirmed after alginate removal (i.e. after microbead dissolution by Ca2+ chelation with 2% w/v Na citrate in IVM medium). In both times, cumulus expansion was visually assessed by stereoscopic microscopy. COC expansion was assessed by cumulus morphology. Indeed, cumuli with continuous edges, consisting of cells in close contact each other, were classified as “compact”, whereas cumuli showing discontinuous edges following cell detachment and production of a viscous extracellular matrix were classified as “expanded”. The percentage of expanded COCs/total COCs was reported. This information has been added in the methods section of the manuscript.

12. Why microbeads of 1000um were the desired size?

Immature COCs from adult ewes are 200-300 μm in diameter. After IVM, COCs reach around 500-600 μm in diameter due to cumulus expansion. For this reason, we planned to produce microbeads of 1000 μm in radius as each of them could suitably house an expanded ovine COC. Moreover, in microbead fabrication experiments, we found that microbeads of radius size closer to 1 (1070.39 ± 88.23 μm) also showed aspect ratio closer to 1 (1.21 ± 0.10), indicating an ideal sphere (Russ, 2011; Tirella et al., 2014; Table 2).

13. Add in the legend of Table 4 the information of needle size (508 μM) and alginate concentration (1%).

This information has been added in Table 4, as requested.

14. Line 331: Include the needle size used for the IVM study.

This information has been added, as requested.

15. Lines 339-342: I would keep the results/table of maturation rates of COCs included in 2% alginate microbeads as supplementary results.

Table 5 has been modified in order to keep the results of COCs in 2% alginate microbeads as supplementary results. A Supplementary file (namely S1 File) has been provided. 

16. Line 409: Change “ithas” by “it has”.

The sentence has been corrected.

17. Lines 418-419: Remove “developed to embryos at the preimplantation stages”, it wasn’t presented in this work.

This part has been removed as data on embryo development were not presented in this work.

18. Would this system applicable for in vitro embryo production?

We have reasons to think that our system would be applicable for in vitro embryo production. For the moment, we analyzed oocyte ability to mature and to reach a suitable bioenergetic/oxidative status (increased expression of TFAM, ATP6 and ATP8 and increased mitochondrial activity and ROS generation ability). Moreover, we analyzed molecular aspects of oocyte cytoplasmic maturation which have been reported as biomarkers of oocyte developmental competence, e.g. the modulation of expression of genes coding for SCMC complex, a protein complex essential biomarker for oocyte meiotic spindle formation and positioning and for zygote progression beyond the first embryonic cell divisions (Li et al., 2008; Lu et al., 2018; Bebbere et al., 2014; 2016; Monk et al., 2017; Qin et al., 2019). Our next aim will be to demonstrate the effectiveness of our 3D IVM system on in vitro embryo production. 

Reviewer #2: In this manuscript, the authors described an optimization of the process of COC-¬laden microbead by using a bioprinter (an automatic generator of hydrogel microspheres). The alginate microspheres produced were characterized in terms of size, shape and stability in culture conditions, using sheep as a large animal model. The results demonstrated high efficiency and reproducibility in the COC encapsulation, in which the cumulus integrity was preserved. After IVM, the 3D system supported oocyte meiotic maturation more efficiently than 2D control. In addition, 3D-matured oocytes exhibited increased expression of genes related to metabolic activation (TFAM, ATP6 and ATP8) and decreased expression of genes related to oocyte competence (KHDC3 and NLRP5) compared to 2D-matured oocytes. Based on the results, they concluded that the proposed method for the production of COC-laden microbeads has high reproducibility and efficiency and that oocyte competence is increased when maturation occurs in the 3D system.

Overall, the manuscript is well organized and the language is appropriate. However, in some parts of the Introduction and Methodology sessions, the text becomes wordy and repetitive.

The main question that has not been answered properly and, therefore, needs more arguments, is the statement and conclusion that the proposed method for 3D maturation improves the oocyte bioenergetic and development potential. Although different expression of genes related to metabolic activation and oocyte competence were observed when comparing oocytes matured in the traditional way (2D) and oocytes matured in 3D, no further evaluation was performed to prove that the difference in the level of the transcripts was actually reflected in different biological responses. For example, mitochondrial function and ATP production/consumption and embryonic development could have been assessed to confirm the increase in metabolic activation and oocyte competence, respectively. Thus, further data would be needed to support the conclusion.

We thank the reviewer for her/his comments and for giving us the opportunity to improve the quality of our manuscript. 

In order to further support the conclusion that the proposed method of 3D maturation improves the oocyte bioenergetic potential, in the revised version, we added new data on effects of the proposed 3D IVM method on oocyte mitochondrial function, expressed in terms of mitochondrial activity, intracellular ROS levels and mitochondria/ROS colocalization after staining with mitochondria- and ROS-specific probes and confocal laser scanning microscopy analysis. This part has been added to the text. New sections for Methods, Results, a new figure (Figure 3) and Discussion sections have been provided. These new data are in agreement with those previously provided on effects of 3D IVM on expression levels of genes coding for TFAM, and the complex V of the respiratory chain (ATP6 and ATP8). Indeed, expression of thes genes has been reported as biomarker of healthy oocyte energy status (Zhao et al., 2014).

Moreover, we analyzed the expression of genes coding for the SCMC protein complex because it is reported as a fundamental biomarker for zygote progression beyond the first embryonic cell divisions (Li et al., 2008; Lu et al., 2018; Bebbere et al., 2014; 2016; Monk et al., 2017; Qin et al., 2019). Our next aim will be to demonstrate the effects of our 3D IVM system on in vitro embryo production.

Specific comments are:

Line 135: cytoplasmic competence was not investigated.

As reported by several studies (Ferreira et al., 2009; Van Blerkom 2008; Sirard 2001; Stojkovic et al., 2001; Davidson 1986), cytoplasmic maturation (or competence) consists of structural (organelle redistribution and cytoskeleton dynamics) and molecular aspects (transcription, storage and processing of maternal mRNA). In the revised version of the manuscript, we have included confocal laser scanning microscopy analysis of the distribution patterns and activity of mitochondria, intracellular ROS localization and mitochondria/ROS colocalization of oocytes matured in the 3D versus the 2D systems. The mitochondria distribution pattern is a parameter of cytoplasmic competence/maturity, as it is reported that in most of mammalian species oocyte mitochondria during IVM move to sites of energy demand such as the pericortical region and the area surrounding the meiotic spindle (Martino et al., 2012). In our experiments, this parameter did not change between methods. However, quantification analyses revealed that mitochondria activity and ROS levels were significantly increased after 3D IVM. Colocalization values remained stable, indicating that observed increased ROS mostly colocalize with mitochondria, thus they are not signal of oxidative stress conditions rather of well-balanced redox status. Data are in agreement with molecular data on TFAM, ATP6 and ATP8 gene expression which can be considered as predictive biomarkers of oocyte mitochondrial function (Zhao et al., 2014). In our opinion, these observations are relevant because cytoplasmic maturation has structural and functional aspects. 

Line 138: For a better understanding of the development of the experiments, information related to the number of COC used, number of COC-laden microbeads cultured and number of structures evaluated in each analysis parameter should be presented in the methodology section, and not in the results. Likewise, the number of replicates in each experiment must also be informed.

This information has been provided. Some information has been provided as methods; other information has been provided as results, because they indicate numbers of tested samples.

Lines 144 and 161: author inform that ovaries from adult sheep (over 1 year) and prepubertal lambs (under 6 months) were recovered for the experiments, but only ovaries from adult cyclic ewes were used?

The sentence has been corrected. Indeed, only ovaries from adult cyclic ewes were used in the present study. 

Line 164: minimum of how many layers?

As reported in previous literature on sheep IVM and IVEP, COCs with a minimum number of 3 cumulus cell layers were selected and included in the study (Varnosfaderani et al., 2020). This information has been included in the revised manuscript. 

Line 177: Have the authors previously tested the supply of oxygen and nutrients in the medium and the removal of waste or is it just an assumption? If it has been tested previously, quote the reference.

The sentence has been modified and two references have been added, as suggested (Cristea et al.,2020; Pandolfi et al., 2017).

Line 229: Did the authors test the COC bioenergetic/oxidative status? These data were not presented.

In the revised version of the manuscript, new data on COC bioenergetic/oxidative status are presented. These data are derived from matured oocytes of some replicates of the 3D versus 2D IVM study which had been stained for mitochondria and ROS. To provide appropriate response to reviewers, stained oocytes underwent confocal laser scanning microscopy analysis and statistics. 

Lines 278-279: microbead stability and microbead morphology were statistically compared between groups, however this analysis was not presented in the results (tables 1 and 4).

We modified the Statistics section and the two tables of morphology (Table 2) and stability (Table 4), as requested.

Line 315, Table 2: when was the analysis performed: immediately after formation or at the end of culture?

Analysis in Table 2 was performed immediately after microbead formation. The title of the table 2 and the results section have been modified for clarity.

Line 320, Table 3: please inform if the evaluation was carried out at the end of the cultures.

COC encapsulation rate was assessed before IVM whereas cumulus espansion rate was analyzed after IVM. The related sentences have been modified in the text, accordingly, for clarity.

Line 354, Table 5: even the highest observed M II rates are still low compared to the available literature. Do the authors have any explanation for this finding?

Our MII rates, even not very high, are comparable with those reported in current literature. In the sheep, significant variations have been reported in IVM efficiency due to intrinsec (donor age and breed, nutrition status and general health conditions) and extrinsec factors (different laboratory procedures; Paramio and Izquierdo, 2014; Zhu et al., 2018).

Line 363: the author can say that there was an increase in the expression of genes related to oocyte mitochondrial activity and embryonic development, but these parameters themselves were not evaluated. More data would be needed to make this statement.

In the revised version of the manuscript, new data on effects of 3D IVM on oocyte bioenergetic/oxidative status have been provided. This paragraph has been modified accordingly.

Line 366: data differ from those reported in the table.

These data are referred to matured oocytes obtained from three additional replicates. These oocytes are different from those reported in table 5. This information has been added in the results, for clarity.

Line 374: please inform the P-value for the tendency.

P-values of the three genes (OOEP1, OOEP2, TLE6) for which no statistically significant differences were observed, have been added.

Line 389: what does the author mean by oocyte competence? Meiotic competence was assessed in this study, but not competence for embryonic development. Please be more specific. Why has embryonic development not been evaluated?

The terms “oocyte competence” have been changed to “oocyte nuclear and cytoplamic maturation”.

Competence for embryonic development was assessed by analyzing the expression of SCMC genes, which are very specific biomarker of oocyte developmental competence (Bebbere et al., 2014). Our sentences on this aspect, have been changed throughout the text, in order to underline the biomarker role of these genes.

Although highly interesting, the culture of embryos was somewhat out of scope with respect to our objectives and will certainly be considered in future studies.

Line 418: was embryo development evaluated?

The authors thank the reviewer for this comment. This sentence was wrong, and it has been modified. 

For the moment, embryo development was not evaluated. It will be part of our next aims. 

Line 422-424: has a standardization study been carried out? If yes, present the data or quote the reference.

By standardized we mean that due to the homogeneous microbead size, it was possible to include numbers of COCs, in each culture well, comparable to those in our conventional 2D system. The sentence has been modified for clarity.

Line 426-427: these variables were not evaluated in the present study.

We modified the sentence by changing the word “guarantee” with the word “try to ensure”.

Line 438: oocyte meiotic competence?

This definition has been corrected, as suggested.

Line 519: unnecessary information, as it is not related to the data presented in the study.

This information has been removed, as requested.

Reviewer #3: GENERAL COMMENTS

The authors devised a one-step automated system to encapsulate sheep cumulus-oocyte complexes (COCs) aiming for a 3D in vitro maturation (IVM) system, as opposed to the traditional 2D system. The 3D system would be a more physiological culture system as it would more closely preserve the natural COC-morphology and, therefore, would improve IVM outcomes. Authors have characterized the size and shape, and reproducibility, of microbeads produced with two different alginate concentrations, and also assessed the microbeads stability during a 24 h in vitro culture. IVM of COCs within the 3D microbeads in comparison to the usual 2D system (control) was evaluated in terms of oocyte nuclear maturation and expression of genes related with oocyte energy metabolism (mitochondrial function) and developmental competence (maternal effect genes related with oocyte competence). Authors determined the parameters for microbeads production which were repeatable and efficient, and stable in culture. Oocyte nuclear maturation increased in 3D culture and transcripts abundance were influenced by the culture system. The authors concluded that the 3D culture system using alginate microbeads was efficient and that it improved oocyte competence. The study is interesting, well designed, bringing new and promising results, and appears to have been appropriately conducted. Overall, the text is well written, organized and clear. There are few corrections, however, regarding some parts of the material and methods (MM) section in which some information is incomplete. Also, some information that should be in the Introduction is in the Discussion and some in the Results should be in MM. Some terms are not so adequate and conclusions are, in part, a little beyond what was presented in the Results, at least considering the way they were written. Therefore, the are suggestions to the text, which are addressed in the Specific comments.

We thank the reviewer for her/his positive comments and for the suggestions which allowed us to improve the quality of the manuscript. 

Specific comments

Title OK

Abstract

L29 – the term COC-laden for the microbeads seems a bit misleading, as laden gives the idea of something heavily loaded with something, and this is not the case as each microbead contains one COC, so the term laden would not be the best in this context. Probably individual COC encapsulated in microbeads or some other term would be better. If changed, do it throughout the text.

This term is commonly used in the literature to mean that gels are loaded with cells or cell structures (1. A combinatorial cell-laden gel microarray for inducing osteogenic differentiation of human mesenchymal stem cells. Alireza Dolatshahi-Pirouz, Mehdi Nikkhah, Akhilesh K. Gaharwar, Basma Hashmi, Enrico Guermani, Hamed Aliabadi, Gulden Camci-Unal, Thomas Ferrante, Morten Foss, Donald E. Ingber & Ali Khademhosseini. Scientific Reports volume 4, Article number: 3896 (2015) 2. Cell-laden alginate dialdehyde–gelatin hydrogels formed in 3D printed sacrificial gel Dalia Dranseikiene, Stefan Schrüfer, Dirk W. Schubert, Supachai Reakasame & Aldo R. Boccaccini Journal of Materials Science: Materials in Medicine volume 31, Article number: 31 (2020)).

However, we do appreciate the Reviewer’s concern that it may lead to a misunderstanding about the number of COCs. Henceforth, for the sake of brevity, we refer to the microbeads containing a COC as "COC-microbeads".

L41 – “… improves oocyte competence…” seems a strong conclusion in relation to the results, as few parameters were assessed (only nuclear maturation and expression of some genes); maybe improves some aspects/parameters of oocyte competence would be better. In the abstract, the conclusion put this way does not seem so much of a problem, as the size of the text may be limited, but this should be addressed, particularly in the final conclusions of the work.

We thank the reviewer and we have modified the abstract and conclusions as suggested. 

Introduction

Some information regarding the genes chosen to be assessed as indicators of cytoplasmic competence should be given here. This information is only given in the Discussion.

Also, the sole expression of some mitochondrial function and maternal effect genes as indicators of cytoplasmic competence, as indicated in the text, does not seem the best description (L135-136); this assessment would be more a molecular competence, although these terms are sometimes used as synonyms; cytoplasmic competence would be best shown as organelle and/or cytoskeleton and/or specific proteins distribution in the cytoplasm and developmental competence would be best addressed using in vitro embryo development (either by IVF or parthenogenetic activation, for example). It would be better to be more specific and indicate that the assessment of the transcripts levels for selected maternal effect and mitochondrial function genes was chosen as an indicator of cytoplasmic or oocyte competence; or clearly indicate references that support that the chosen transcripts are directly related with and are markers of oocyte developmental competence.

As mentioned, the Introduction does not consider these factors – mitochondrial function and maternal effect genes – so it does not make clear the relation between cytoplasmic maturation and the assessments made in the experiments that were carried out.

We thank the reviewer for these suggestions.

We modified the Introduction, by shortening it, as also suggested by reviewer 1, and by moving from the Discussion some information concerning genes chosen as indicators of oocyte bioenergetic and developmental competence. We indicated references of papers reporting that these genes are biomarkers of matured oocyte bioenergetic and developmental potential, respectively.

Material and Methods

Part of the figures in Results include images of the setup of the system, and these parts of the figures could be placed in the MM. Sometimes only by the text it is difficult to understand how the microbeads were prepared, and mostly how the COCs individually were placed inside the microbeads. Therefore, the images referring to this, which are in Results, could be placed in MM to make it easier to understand the method.

We moved Figure 1 to the MM section, in the paragraph “Alginate microbead fabrication”, as suggested.

L151 and L160 – As a suggestion, invert IVM medium and COC retrieval

These two paragraphs have been inverted, as suggested.

L171-172 – are these notations correct – 5 divided by 20 and 1 divided by 100, and both between parenthesis?

The notations indicated respectively the range of flow rate and pressure. We changed the notations to (5-20 ± 1) μLs−1 and (1-100 ± 0.4) kPa.

L173 “…actuated…” would this be activated?

We rephrased line 173 to “as the extrusion system is activated”, for better clarity.

L176 – indicate here between parenthesis the range of desired dimensions

We have added this information.

177 “…as well as…”

We have corrected the sentence.

L185 – same as in L173

We modified “actuated” with “activated”.

L205 – laden, could be changed to another term

We changed with COC-microbead.

L215 – here is when we finally get to know that there is one COC/microbead; it is not so easy to understand how, so referring to a figure here would be helpful.

As, also requested by reviewer 1, Figure 2 has been moved to the MM section, paragraph “Alginate COC-microbead generation”.

L221-222 – how many microbeads with COCs inside in each well?

This information has been added.

L226 – how many COCs/well?

This information has been added.

L227 – time and temperature for chelation of calcium?

This information was added.

L229 – which bioenergetics/oxidative status? This was not measured, you did not analyse ROS levels or ATP levels or things like that, be more specific of what you´ve actually done

In the revised version of the manuscript, new data on oocyte bioenergetic/oxidative status have been added.

L229-232 No need to say “Those COCs destined for … and those destined for… underwent cumulus separation…” as all them were denuded. Just say that at the end of IVM COCs were denuded for analysis by incubation and so on, otherwise the sentence is repetitive.

Thank you, the sentence has been modified.

L235 “… chromatin, oocytes…”

The sentence has been corrected

L245 – total RNA of how many oocytes / sample? What was the size of the oocyte pool? How many pools were used?

This information has been added.

L257 – “The list of analysed genes is reported in Table 1” can be removed as the next sentence mentions the oligonucleotide primers and Table 1, so it is repetitive (or remove the Table 1 in parenthesis).

This part has been corrected, as suggested.

L276-277 the analysis of cumulus expansion rate is mentioned as analysed, but this analysis is not described; this description should be included in MM.

Description on COC expansion analysis has been included in MM, as requested.

L279 morphology of the microbeads was also analysed, but no description of how this was done is provided; this should be included in MM.

Microbead morphology was assessed by evaluating their dimensions (radius) and the shape descriptor “aspect ratio” (according to Russ et al., 2011), as reported by Tyrella et al., 2014. The sentence, in the paragraph “Microbeads morphological characterization and stability” has been modified for clarity.

Results

L291 – define “aspect ratio” and the meaning of being ≈1. Actually, a description of how the shape of the beads were analysed should be detailed in the MM

The requested definitions have been added in the results section. The description of how the shape of the beads was analysed has been detailed in MM, as requested.

L294-298 – COC encapsulation rate and cumulus expansion rate – these should be explained how they were measured in MM; is there a difference between encapsulation efficiency, rate and levels? Specify what exactly each term means as this part is a bit confusing as it is.

The sentence has been modified, for clarity. The terms “encapsulation efficiency” and “cumulus expansion efficiency” have been better clarified in MM. The “encapsulation efficiency” was measured before IVM as “encapsulation rate” which was the proportion of COCs uploaded in the Sphyga syringe which came out as included in microbeads in a 1/1 proportion. Those COCs which resulted non-included in microbeads or included in a more than 1/1 proportion (in groups of 2 or more COCs) were excluded as they represent the “No” response. The “cumulus expansion efficiency” was measured after IVM as “cumulus expansion rate” which was the proportion of encapsulated COCs whose cumulus expanded after culture. Those COCs which remained compact after culture were excluded as they represent the “No” response.

L317 – as mentioned, the assessment of shape should be detailed in MM

The assessment of microbead shape has been further detailed in MM, as requested.

L334 – cumulus expansion evaluation should be explained in MM

The cumulus expansion evaluation method has been further explained in MM, as requested.

L338-339 – define what was considered abnormal chromatin configuration and activated oocytes in MM

In the MM section we have now included criteria for abnormal chromatin configuration and activated oocytes, as requested. 

Oocytes showing either multipolar meiotic spindle or irregular chromatin clumps or no chromatin were considered as abnormal (Dell’Aquila et al., 2003). Oocytes with the first polar body extruded and at least one pronucleus as a sign of spontaneous parthenogenetic activation were indicated as activated. These sentences have been added in the MM section. 

As requested by reviewer 1 (point n.15), we removed the line of oocytes matured in 2% alginate microbeads which had activated oocytes. These data are now reported in Supplementary file (S1 File). 

No effects of 3D vs 2D IVM systems were noticed on the incidence of these chromatin configurations.

L363 – “… improves oocyte energy and developmental potential” – actually, not quite this. Oocyte energy was not assessed, if so, there should be some measure of this parameter either by evaluating ATP production, mitochondrial activity or potential by specific staining, or other methods. You analysed the levels of transcripts for genes related with mitochondrial function, not directly oocyte energy. Otherwise, provide a reference that indicates that the expression of chosen genes can be directly linked to a cell´s energy levels. As there is nothing of this in the Introduction it is hard to find the direct connection authors are trying to make here. The same goes for developmental potential; no embryo production was evaluated. Again, the assessment was on transcript levels for maternal effect genes; could include reference indicating that expression of such genes are directly linked to developmental potential in vivo and/or in vitro; also here, no prior information on the genes is given in the Introduction. There was an effect on the transcript levels for genes related to mitochondrial function and maternal effect, which can affect developmental potential; this is an indirect assessment.

In the revised version of the manuscript, data of mitochondria distribution pattern and activity, intracellular ROS localization and mitochondria/ROS colocalization of oocytes matured in the 3D versus the 2D systems, stained with specific probes and analyzed under confocal laser scanning microscopy were included. In our experiments, mitochondrial activity and ROS levels were significantly increased after 3D IVM. These observations are in agreement with our findings of increased relative abundance of TFAM, ATP6 and ATP8 mRNAs. Thus, the expression of these genes can be linked to cell energy levels (Zhao et al., 2014). We added this information in the Introduction in order to make appropriate connection, as suggested.

Concerning embryo developmental potential, unfortunately no embryo production was performed in this study. However, as suggested by the reviewer, we reinforced the concepts that assessment of “oocyte developmental potential” was based on particularly specific transcript levels for maternal effect genes which are predictive biomarkers of the zygote’s ability to undertake first cleavage stages and included all references indicating that expression of such genes are directly linked to developmental potential in vivo and/or in vitro (Bebbere et al., 2014). Also here, information on these genes has been added in the Introduction, as suggested.

L366-367 – how were the 77 and 72 oocytes pooled in groups of 25 oocytes? this information should be in MM

This information has been added.

L370 – what is SCMC? This is the first time in the manuscript this is mentioned and it only cited; then, it is only explained in the Discussion. This should be addressed in the Introduction or at least in the MM to justify the choice of genes to be studied (same for mitochondrial genes) and why they can be used as indicators of oocyte competence.

The role of the analyzed genes has been addressed in the Introduction. 

L374 – if there was a tendency what was the value of P?

Requested P-values have been included.

L377 – it is not quite transcriptional activity; this is different from transcripts abundance or relative levels; probably best to use transcript abundance for genes related with mitochondrial function and maternal effects

We thank the reviewer and we changed the title of the figure (now “Fig 4), as suggested.

Discussion

L389-390 – this part of the text mentions assessment of oocyte competence by cytogenetic and molecular analysis. Oocyte competence was not measured, but some parameters related to that, and cytogenetic analyses were not performed (the chromosomes were not studied, only chromatin configuration as indication of meiosis stage); nuclear maturation is not a cytogenetic analysis it is a cytological one. Attention with the use of some terms.

We agree with the reviewer and we thank her/him for this correction. Actually, we performed a “cytological analysis”. Indeed, mammalian oocyte nuclei exhibit characteristic chromatin configurations, which are subject to dynamic modifications through oogenesis and meiotic maturation. During meiotic arrest at the diplotene stage, the chromosomes form a loose chromatin mass. The decondensed configuration of chromatin then undergoes profound rearrangements leading to formation of groups of chromosomes which can be seen as distributed on metaphase plates or migrating through meiotic spindle fibers. Analysis of these configurations are “cytological” and not “cytogenetic” analysis which are performed on single chromosomes. This part has been modified.

L418-419 – the way this part is written leads to the understanding that you have obtained the development of embryos; I understand that your results corroborate previous findings that showed such embryo development, but the phrasing is dubious, so it would be best to rephrase this part to make clear what exactly you are trying to inform.

The sentence has been rephrased, as requested. 

L423 – this information is missing in MM

This information has been added also in MM.

L438-443 – how does your MII rate compare to other 2D studies in sheep IVM? Is it similar to what was found in your 2D control system? And the rate of anomalies?

Our MII rates, even not very high, are comparable with current available literature. In the sheep, significat variations are reported in IVM efficiency due to intrinsic (donor age and breed, nutrition status and general health conditions) and extrinsic factors (different laboratory procedures, such as. i.e. serum supplement, hormones, antibiotics etc. (Paramio and Izquierdo, 2014; Zhu et al., 2018)

L448-459 – this entire part is more like a review explaining the genes that were studied; this could be (shorter even) in the Introduction and/or MM. Starting from the end of L459 is more like a discussion. Reorganize and shorten this part.

This part has been reorganized and shortened, as suggested.

L471 – include “…could potentially improve oocyte mitochondria function” or provide reference that the increase in transcripts for these genes is a marker of enhanced mitochondrial function and make this information clear.

The sentence has been modified and supporting references have been added, as suggested. 

L472-512 this part mentions and then explains SCMC; this (also can be shorter) should be in the Introduction and/or MM (L370 just cites SCMC); the explanation on the genes goes up to L486; from here on in the text is more like a discussion contrasting your findings with their possible meaning for oocyte function. Reorganize and shorten this part.

This part has been reorganized and shortened, as suggested.

L492- 493 you found accelerated degradation of maternal transcripts? Did you do kinetics of transcripts levels during IVM? What you found was that transcripts were decreased in 3D oocytes compared to 2D; rephrase this part to reflect what was observed.

This part has been rephrased to reflect what was observed and the correlation between our data and those reported by Li et al., 2008 and Bebbere et al., 2014.

L493 “…similar to the, as relationship…” it appears that there is some word or part missing after similar to the, is similar to the what?

The reviewer is right. The sentence had some missing parts and has been rephrased.

L511 – “… potentially improved oocyte competence” or “… improved some parameters related with oocyte competence”, as assessment of competence was limited. It would be quite interesting in the future to evaluate effects on embryo development and quality and other functional parameters in oocytes and cumulus cells.

The sentence has been rephrased, as indicated.

L516 - the conclusion oversteps a bit what was observed in the results, what was improved was nuclear maturation and the expression of some mitochondrial function and maternal effect genes; or make it clear throughout the text, with literature support, that these assessments (transcripts levels of chosen genes) can be used as reliable indicators of bioenergetic and developmental potential in oocytes. More studies are needed to further address the effects of the novel culture system on COC function, which seem to be, anyway, quite promising. 

The conlusions have been modified in order to be more related to what is described in the results.

6. PLOS authors have the option to publish the peer review history of their article (what does this mean?). If published, this will include your full peer review and any attached files.

Do you want your identity to be public for this peer review? For information about this choice, including consent withdrawal, please see our Privacy Policy.

Reviewer #1: No

Reviewer #2: No

Reviewer #3: No

---

## [Decision Letter · Decision Letter 1]

2 Jul 2020

PONE-D-20-09540R1

One-step automated bioprinting-based  method for cumulus-oocyte complex microencapsulation  for 3D in vitromaturation

PLOS ONE

Dear Dr. Mastrorocco,

Thank you for submitting your manuscript to PLOS ONE. After careful consideration, we feel that it has merit but does not fully meet PLOS ONE’s publication criteria as it currently stands. Therefore, we invite you to submit a revised version of the manuscript that addresses the points raised during the review process.

Please authors, carefully consider each of the questions raised by reviewers # 1 and # 2.

We look forward to receiving your revised manuscript.

Kind regards,

Marcelo Fábio Gouveia Nogueira, Associate Professor, Ph. D.

Academic Editor

PLOS ONE

Reviewers' comments:

Reviewer's Responses to Questions

**Comments to the Author**

1. If the authors have adequately addressed your comments raised in a previous round of review and you feel that this manuscript is now acceptable for publication, you may indicate that here to bypass the “Comments to the Author” section, enter your conflict of interest statement in the “Confidential to Editor” section, and submit your "Accept" recommendation.

Reviewer #1: All comments have been addressed

Reviewer #2: All comments have been addressed

Reviewer #3: All comments have been addressed

2. Is the manuscript technically sound, and do the data support the conclusions?

Reviewer #1: Yes

Reviewer #2: Yes

Reviewer #3: Yes

3. Has the statistical analysis been performed appropriately and rigorously? 

Reviewer #1: I Don't Know

Reviewer #2: Yes

Reviewer #3: Yes

4. Have the authors made all data underlying the findings in their manuscript fully available?

Reviewer #1: Yes

Reviewer #2: (No Response)

Reviewer #3: Yes

5. Is the manuscript presented in an intelligible fashion and written in standard English?

Reviewer #1: Yes

Reviewer #2: Yes

Reviewer #3: Yes

6. Review Comments to the Author

Reviewer #1: The article titled “One-step automated bioprinting-based method for cumulus-oocyte complex microencapsulation for 3D in vitro maturation” has been substantially improved after the author’s modification. The text is more reader friendly, the aim of the manuscript is clearer, and some information, previously missing, has been added in the text. I am satisfied with the answers addressed to my earlier comments.

In this new submission, the author’s included new analysis to assess the bioenergetic/oxidative status of the oocytes aimed to make cleared the effect of the 3D IVM system in the mitochondria activity, intracellular ROS levels, and mitochondria/ROS colocalization.

Theses analyses corroborates to the data presented earlier, however the discussion of these results need to be improved. The lines 484-486 of the discussion session need reference. And the effect of higher mitochondria activity and ROS levels in the oocyte from 3D-IVM system need to be discussed.

Specific comments:

Line 37: Change “that” by “than”.

Line 251-252: The definition of spontaneous parthenogenetic activation should be reviewed. Moreover, this classification of activated oocytes was not presented in the results.

Lines 371-374: the information in these lines is conflicting. First it says there was a significant difference between 3D and 2D groups in relation to oocyte abnormal chromatin configuration. Followed by a sentence that says: no effect of different systems was noticed on the incidence of specific abnormal chromatin configurations. In figure 5, it shows a difference between 3D and 2D groups in relation to oocyte abnormal chromatin configuration. Please, confirm the information in these lines.

Lines 404-405: Remove the sentence “Oocytes matured in 3D IVM showed significantly increased mitochondrial activity and intracellular ROS levels.” This is not information for a legend.

Lines 413-414: Remove the sentence “Increased MitoTracker (C1 versus C2) and DCF fluorescent intensity (D1 versus D2) are visible in 3D IVM oocytes.” This is not information for a legend.

Line 416: I would consider changing “improves oocyte molecular maturation” by “modulates oocyte gene expression”. The term “molecular maturation” refers to many other processes and transcripts that were not investigated in this study.

Line 427: Please, consider to change the expression “tended to decrease” by “was similar between groups”. The p-values informed are not consistent with a classification as tendency.

Line 511: Remove the words “reduction (although not significant) of”

Lines 511-514: I do not understand the comparison between the data from the present study with the GDF-9 quantification analysis by Morselli et al. [34].

Reviewer #2: All the questions raised were appropriately answered by the authors and the changes were incorporated into the text, making it suitable for publication. The only observation I still make is the micrograph representing the chromosomal configuration of the 2D group in Figure 3, as apparently the oocyte has not been fully denuded and the chromatin of the cumulus cells can hinder the visualization of the oocyte chromatin, leading to errors in the interpretation of the nuclear configuration of the oocyte. oocyte. Therefore, this image should be replaced by one of better representation.

Reviewer #3: The authors have included new experiments and data, that together with corrections made to the manscript, have improved the work. Questions raised previously were apporpriately addressed.

7. PLOS authors have the option to publish the peer review history of their article (what does this mean?). If published, this will include your full peer review and any attached files.

Reviewer #1: No

Reviewer #2: No

Reviewer #3: No

---

## [Author Response · Author response to Decision Letter 1]

7 Aug 2020

PONE-D-20-09540R1

One-step automated bioprinting-based method for cumulus-oocyte complex microencapsulation for 3D in vitro maturation

PLOS ONE

Dear Dr. Mastrorocco,

Thank you for submitting your manuscript to PLOS ONE. After careful consideration, we feel that it has merit but does not fully meet PLOS ONE’s publication criteria as it currently stands. Therefore, we invite you to submit a revised version of the manuscript that addresses the points raised during the review process.

Please authors, carefully consider each of the questions raised by reviewers # 1 and # 2.

We look forward to receiving your revised manuscript.

Kind regards,

Marcelo Fábio Gouveia Nogueira, Associate Professor, Ph. D.

Academic Editor

PLOS ONE

Reviewers' comments:

Reviewer's Responses to Questions

Comments to the Author

1. If the authors have adequately addressed your comments raised in a previous round of review and you feel that this manuscript is now acceptable for publication, you may indicate that here to bypass the “Comments to the Author” section, enter your conflict of interest statement in the “Confidential to Editor” section, and submit your "Accept" recommendation.

Reviewer #1: All comments have been addressed

Reviewer #2: All comments have been addressed

Reviewer #3: All comments have been addressed

2. Is the manuscript technically sound, and do the data support the conclusions?

Reviewer #1: Yes

Reviewer #2: Yes

Reviewer #3: Yes

3. Has the statistical analysis been performed appropriately and rigorously?

Reviewer #1: I Don't Know

Reviewer #2: Yes

Reviewer #3: Yes

4. Have the authors made all data underlying the findings in their manuscript fully available?

Reviewer #1: Yes

Reviewer #2: (No Response)

Reviewer #3: Yes

5. Is the manuscript presented in an intelligible fashion and written in standard English?

Reviewer #1: Yes

Reviewer #2: Yes

Reviewer #3: Yes

6. Review Comments to the Author

Reviewer #1: The article titled “One-step automated bioprinting-based method for cumulus-oocyte complex microencapsulation for 3D in vitro maturation” has been substantially improved after the author’s modification. The text is more reader friendly, the aim of the manuscript is clearer, and some information, previously missing, has been added in the text. I am satisfied with the answers addressed to my earlier comments.

In this new submission, the author’s included new analysis to assess the bioenergetic/oxidative status of the oocytes aimed to make cleared the effect of the 3D IVM system in the mitochondria activity, intracellular ROS levels, and mitochondria/ROS colocalization.

Theses analyses corroborates to the data presented earlier, however the discussion of these results need to be improved. The lines 484-486 of the discussion session need reference. And the effect of higher mitochondria activity and ROS levels in the oocyte from 3D-IVM system need to be discussed.

We thank the reviewer for giving us the opportunity to improve the quality of our manuscript. The discussion on results on the effects of 3D IVM on mitochondria activity and ROS levels has been improved, as requested. References for lines 484-486 have been provided. Possible explanations of higher mitochondria activity and ROS levels found in matured oocytes from the 3D-IVM system have been provided.

Specific comments:

Line 37: Change “that” by “than”.

The word has been corrected.

Line 251-252: The definition of spontaneous parthenogenetic activation should be reviewed. Moreover, this classification of activated oocytes was not presented in the results.

Oocyte spontaneous parthenogenetic activation is a post-Metaphase II condition in which chromatin de-condensation and pronuclear formation take place in the absence of sperm penetration and fertilization. This event is related to the decline in the activity of the two major kinases that regulate meiotic cycle (MPF and MAPK; Ariu et al., 2014). In physiological conditions, MPF and MAPK decline occur in association with fertilization in order to allow male (sperm) and female (oocyte Metaphase II) chromatin de-condensation to obtain the two pronuclei. In experimental conditions, parthenogenetic activation is used to test oocyte pronuclear formation ability, thus its developmental competence (Schoevers et al., 2016) and it is frequently used in sheep oocytes and other animal models. In most of dated and recent references, parthenogenetically activated oocytes are defined as those oocytes showing one pronucleus (Ledda et al., 1997; Sun et al., 2002; Jiang et al., 2014), but the first polar body is extruded (See Figure 3D in Ariu et al., 2014). For these reasons, we think that our definition in the Methods section section is correct. However, upon reviewer suggestion, we changed the sentence for clarity and provided a supporting references (Ariu et al., 2014) 

The classification of activated oocytes was presented in Supplementary file 1 because oocytes with this chromatin configuration were observed only after IVM in 2% alginate microbeads. A possible explanation for increased rates of activated oocytes in 2% vs 1% or vs control bidimensional IVM could be related to the fact that COCs cultured in microbeads with too high alginate concentration, impeding cumulus expansion, may have undergone dysmetabolic conditions, thus their MPF/MAPK levels could have been reduced, with consequent MII chromatin de-condensation and pronucleus formation.

Lines 371-374: the information in these lines is conflicting. First it says there was a significant difference between 3D and 2D groups in relation to oocyte abnormal chromatin configuration. Followed by a sentence that says: no effect of different systems was noticed on the incidence of specific abnormal chromatin configurations. In figure 5, it shows a difference between 3D and 2D groups in relation to oocyte abnormal chromatin configuration. Please, confirm the information in these lines.

Certainly the reviewer intended to refer to table 5. We confirm the information that the 3D system reduced the incidence of oocyte chromosomal abnormalities. The conflicting sentence has been eliminated, as suggested. 

Lines 404-405: Remove the sentence “Oocytes matured in 3D IVM showed significantly increased mitochondrial activity and intracellular ROS levels.” This is not information for a legend.

This sentence has been removed, as requested.

Lines 413-414: Remove the sentence “Increased MitoTracker (C1 versus C2) and DCF fluorescent intensity (D1 versus D2) are visible in 3D IVM oocytes.” This is not information for a legend.

This sentence has been removed, as requested.

Line 416: I would consider changing “improves oocyte molecular maturation” by “modulates oocyte gene expression”. The term “molecular maturation” refers to many other processes and transcripts that were not investigated in this study.

This sentence has been modified, as suggested.

Line 427: Please, consider to change the expression “tended to decrease” by “was similar between groups”. The p-values informed are not consistent with a classification as tendency.

This sentence has been modified, as suggested.

Line 511: Remove the words “reduction (although not significant) of”

The suggested worlds have been removed.

Lines 511-514: I do not understand the comparison between the data from the present study with the GDF-9 quantification analysis by Morselli et al. [34].

In our opinion, our results on down-regulation of expression of genes coding for SCMC complex in 3D IVM are in agreement with those on down-regulated expression of GDF-9 observed in canine after 3D IVM (Morselli et al. [34]). Indeed, similarly to SCMC, GDF-9 is involved in oocyte developmental competence and its expression is reported to decrease during in vivo and in vitro oocyte maturation (De Los Reyes et al., Theriogenology 2013). 

Reviewer #2: All the questions raised were appropriately answered by the authors and the changes were incorporated into the text, making it suitable for publication. The only observation I still make is the micrograph representing the chromosomal configuration of the 2D group in Figure 3, as apparently the oocyte has not been fully denuded and the chromatin nof the cumulus cells can hinder the visualization of the oocyte chromatin, leading to errors in the interpretation of the nuclear configuration of the oocyte. oocyte. Therefore, this image should be replaced by one of better representation.

We thank the reviewer for giving us the opportunity to improve the quality of our manuscript. As requested, in Figure 3, the micrograph representing the chromosomal configuration of an oocyte cultured in the 2D system has been replaced with a better one without cumulus cells in order to improve chromatin visualization. 

Reviewer #3: The authors have included new experiments and data, that together with corrections made to the manuscript, have improved the work. Questions raised previously were appropriately addressed.

We thank reviewer 3 for these positive comments. 

7. PLOS authors have the option to publish the peer review history of their article (what does this mean?). If published, this will include your full peer review and any attached files.

Do you want your identity to be public for this peer review? For information about this choice, including consent withdrawal, please see our Privacy Policy.

Reviewer #1: No

Reviewer #2: No

Reviewer #3: No

---

## [Decision Letter · Decision Letter 2]

25 Aug 2020

One-step automated bioprinting-based method for cumulus-oocyte complex microencapsulation for 3D in vitro maturation

PONE-D-20-09540R2

Dear Dr. Mastrorocco,

We’re pleased to inform you that your manuscript has been judged scientifically suitable for publication and will be formally accepted for publication once it meets all outstanding technical requirements.

Kind regards,

Marcelo Fábio Gouveia Nogueira, Associate Professor, Ph. D.

Academic Editor

PLOS ONE

Additional Editor Comments (optional):

Reviewers' comments:

Reviewer's Responses to Questions

**Comments to the Author**

1. If the authors have adequately addressed your comments raised in a previous round of review and you feel that this manuscript is now acceptable for publication, you may indicate that here to bypass the “Comments to the Author” section, enter your conflict of interest statement in the “Confidential to Editor” section, and submit your "Accept" recommendation.

Reviewer #1: All comments have been addressed

Reviewer #2: All comments have been addressed

2. Is the manuscript technically sound, and do the data support the conclusions?

Reviewer #1: Yes

Reviewer #2: Yes

3. Has the statistical analysis been performed appropriately and rigorously? 

Reviewer #1: Yes

Reviewer #2: Yes

4. Have the authors made all data underlying the findings in their manuscript fully available?

Reviewer #1: Yes

Reviewer #2: Yes

5. Is the manuscript presented in an intelligible fashion and written in standard English?

Reviewer #1: Yes

Reviewer #2: Yes

6. Review Comments to the Author

Reviewer #1: The authors answered to all the remarks and questions appropriately. Changes to the manuscript have been made accordingly. All together have improved the work.

Reviewer #2: The micrograph representing the chromosomal configuration of the oocyte cultured in the 2D system has been replaced by a better one. All questions raised previously were appropriately addressed.

7. PLOS authors have the option to publish the peer review history of their article (what does this mean?). If published, this will include your full peer review and any attached files.

Reviewer #1: No

Reviewer #2: No

---

## [Editor Report · Acceptance letter]

1 Sep 2020

PONE-D-20-09540R2 

One-step automated bioprinting-based method for cumulus-oocyte complex microencapsulation for 3D in vitro maturation 

Dear Dr. Mastrorocco:

I'm pleased to inform you that your manuscript has been deemed suitable for publication in PLOS ONE. Congratulations! Your manuscript is now with our production department. 

Kind regards, 

on behalf of

Dr. Marcelo Fábio Gouveia Nogueira 

Academic Editor

PLOS ONE